

# Syn-kinematic hydration reactions, dissolution-precipitation creep and grain boundary sliding in experimentally deformed plagioclase - pyroxene mixtures

Sina Marti[1], Holger Stünitz[2,3], Renée Heilbronner[1], Oliver Plümper[4], and Rüdiger Kilian[1]

[1]Department of Environmental Sciences, Basel University, Switzerland
[2]Department of Geosciences, UiT the Arctic University of Norway, Norway
[3]Institut des Sciences de la Terre d'Orléans (ISTO), Université d'Orléans, France
[4]Department of Earth Sciences, Utrecht University, The Netherlands

**Correspondence:** Sina Marti (sina.marti@ed.ac.uk)

**Abstract.** While it is widely observed that mafic rocks are able to exeprience high strains by viscous flow, details on their rheology and deformation mechanisms are poorly constrained. Here, rock deformation experiments on four different, water-added plagioclase-pyroxene mixtures are presented: (i) plagioclase(An60-70) – clinopyroxene – orthopyroxene, (ii) plagioclase(An60) – diopside, (iii) plagioclase(An60) – enstatite and (iv) plagioclase(An01) – enstatite. Samples were deformed in general shear at strain rates of $3 \cdot 10^{-5}$ to $3 \cdot 10^{-6}$ s$^{-1}$, 800 °C and confining pressure of 1.0 or 1.5 GPa. Results indicate that dissolution-precipitation creep (DPC) and grain boundary sliding (GBS) are the dominant deformation mechanisms. Coinciding with sample deformation, syn-kinematic mineral reactions yield abundant nucleation of new grains; the resulting intense grain size reduction is considered crucial for the activity of DPC and GBS. In high strain zones dominated by plagioclase, a weak, non-random and geometrically consistent crystallographic preferred orientation (CPO) is observed. Usually, a CPO is considered a consequence of dislocation creep, but the experiments presented here demonstrate that a CPO can develop during DPC and GBS. This study provides new evidence for the importance of DPC and GBS in mid-crustal shear zones within mafic rocks, which has important implications on understanding and modelling of mid-crustal rheology and flow.

## 1   Introduction

Viscous deformation of crustal rocks is usually dominated either by intra-crystalline deformation (dislocation creep), or by mechanisms such as diffusion creep and grain boundary sliding. Apart from being rate and temperature sensitive, the rheology of viscously deforming rocks is also observed to be material dependent (for a comprehensive list of flow law parameters for different rock types see e.g., Kohlstedt et al. (1995); Shaocheng and Bin (2002); Bürgmann and Dresen (2008); Burov (2011) and references therein). Flow laws for viscous creep exist for different types of rocks, with the majority of these flow laws being determined for monomineralic materials. It is often suggested that viscous deformation in monomineralic aggregates at



mid- to lower crustal conditions is dominated by dislocation creep. Grain growth in monomineralic aggregates at the elevated temperatures of the mid- to lower crust is assumed to be extensive and the resulting large grain size is expected to render diffusion creep less efficient than dislocation creep. Insights into deformation mechanisms, slip systems, and flow law parameters have been obtained from experimental studies, e.g., for plagioclase: Tullis and Yund (1985); Shaocheng and Mainprice (1987);

Tullis and Yund (1991); Dimanov et al. (1999); Rybacki and Dresen (2000); Stünitz and Tullis (2001); Stünitz et al. (2003); Ji et al. (2004); Barreiro et al. (2007), and for pyroxene: Lallemant (1978); Kolle and Blacic (1982); Raterron and Jaoul (1991); Mauler et al. (2000); Bystricky and Mackwell (2001); Hier-Majumder et al. (2005); Chen et al. (2006); Zhang et al. (2006). For polyphase mixtures of gabbroic composition, data from high-temperature deformation experiments are published by Dimanov et al. (2003, 2007); Dimanov and Dresen (2005). Depending on the grain size, the differential stress and the volume

fraction of pyroxene (as the stronger phase in their pyroxene-plagioclase mixtures), the dominant deformation mechanism identified by these authors is either diffusion creep or dislocation creep. The strain rate of the two-phase aggregates thereby is suggested to be a combination of the strain rates of the individual phases e.g., Dimanov et al. (2003). No mineral reactions were observed in these experiments.

In somewhat lower-temperature experiments and mostly under hydrous conditions, Rutter et al. (1985); Getsinger and Hirth

(2014), and Stünitz and Tullis (2001) performed deformation experiments with syn-kinematic hydration reactions. Phase mixing was found to be (partly) due to the nucleation of new phases. The authors suggest that the dominant deformation mechanism is grain size sensitive creep by a mix of diffusion creep and grain boundary sliding. Rutter et al. (1985) state explicitly that they interpret diffusion creep in the sense of dissolution-precipitation creep.

In polymineralic mixtures, several processes are known to influence the deformability and the dominating deformation mech-

anism of the bulk aggregate. The occurrence of mineral reactions and nucleation causes grain size reduction (e.g., Brodie and Rutter, 1987; Fitz Gerald and Stünitz, 1993; Newman et al., 1999; de Ronde et al., 2005) and can lead to (further) phase mixing. Grain pinning due to secondary phases will impede grain growth (e.g., Olgaard and Evans, 1986; Berger and Herwegh, 2004; Linckens et al., 2011), and diffusion is expected to be faster along phase boundaries compared to grain boundaries (e.g. Hickman and Evans, 1991; Wheeler, 1992; Sundberg and Cooper, 2008). These factors enhance diffusion creep rates and thus

may lead to a switch of the dominant deformation mechanism from dislocation creep in monomineralic layers, to diffusion creep in polymineralic layers (Etheridge and Wilkie, 1979; Mehl and Hirth, 2008; Linckens et al., 2011; Kilian et al., 2011).

In the absence of fluids, metastable mineral assemblages can be preserved over long time periods (e.g., Jamtveit et al., 2016). When fluids infiltrate, mineral reactions take place. Under deviatoric stress conditions, deformation is frequently localized along these zones of fluid infiltration and metamorphic reactions (e.g., Austrheim, 1987). A positive feedback between defor-

mation and metamorphic reactions has been recognized for some time but the exact mechanisms of the interaction are still not sufficiently understood. The positive feedback with metamorphic reactions may not be the same for all deformation mechanisms and thus the syn-kinematic occurrence of mineral reactions is a factor that can influence the dominance of a certain deformation mechanism.

In the presence of fluids, mafic rocks are particularly susceptible to reactions during changing temperatures and pressures,

representing a suitable material to study the interplay between reaction and deformation. That high strain zones such as ul-





tramylonites usually consist of a phase mixture, indicates their ability to deform at higher strain rates (or lower stresses) than monomineralic aggregates and emphasizes their importance for localizing deformation.

In this study, we present results from deformation experiments on water-added plagioclase-pyroxene mixtures. At the imposed pressure-temperature conditions of $\sim 1.0 - 1.5$ GPa and 800 °C, deformation takes place within the lower temperature range

of the viscous regime. The metastability of the starting material in the $H_2O$-present system causes syn-kinematic mineral reactions, thus facilitating the interplay between reaction and deformation in the experiments.

## 2 Materials and Methods

### 2.1 Experiments

#### 2.1.1 Starting materials

Experiments are performed on five different starting materials (composition of starting material and chemical composition of minerals are given in Table 1; mineral abbreviations after Whitney and Evans (2010)):

(i) MD: Crushed Maryland Diabase, (Kronenberg and Shelton, 1980; Marti et al., 2017), using a grain size fraction $\leq 125$ $\mu$m. The Maryland Diabase starting material has a modal composition of plagioclase: $\sim 57$ vol%, Clinopyroxene: $\sim 32$ vol%, Orthopyroxene: $\sim 8$ vol%, Accessories: $\sim 3$ vol% (Qz, Kfs, Ilm, Mag, Bt, Ap).

(ii) An60+En: Synthetic mixture of Sonora labradorite ($\sim$ An60) and Damaping enstatite. powder. Grain size fraction of $\sim$ 2 - 125 $\mu$m, and 40 - 180 $\mu$m.

(iii) An60+Di: Synthetic mixture of Sonora Labradorite and Damaping diopside (D) powder. Grain size fraction of $\sim$ 2 - 125 $\mu$m.

(iv) An60+Di: Synthetic mixture of Sonora labradorite and Cranberry Lake diopside powder. Grain size fraction of $\sim$ 2 - 125 75 $\mu$m, and 40 - 125 $\mu$m.

(v) Ab+En: Synthetic mixture of Alpe Rischuna albite ($\sim$ Ab98) and Damaping enstatite powder. Grain size fraction of $\leq$ 125 $\mu$m, and 40 - 180 $\mu$m.

Detailed description of the sample preparation can be found in the Appendix. Synthetic plagioclase-pyroxene powders are mixed with phase proportions of $\sim 57$ vol% plagioclase to 43 vol% pyroxene. Either 0.2 $\mu$l (0.18 wt%) or 0.12 $\mu$l (0.11 wt%)

$H_2O$ is added to the sample.

#### 2.1.2 Experimental conditions and sample assembly

Experiments are performed using the Griggs-type deformation apparatus at the University of Tromso, Norway. Experiments are run at confining pressures (Pc) of $\sim 1.0$ and 1.5 GPa, temperatures (T) of 800 °C and (axial) displacement rates of $\sim$



$2 \cdot 10^{-8}$ to $2 \cdot 10^{-9}$ ms$^{-1}$ resulting in bulk strain rates of $\sim 3 \cdot 10^{-5}$ to $3 \cdot 10^{-6}$ s$^{-1}$. General shear type of flow is achieved by placing

the rock powder (0.11 g) between cylindrical alumina forcing blocks (diameter of 6.33 mm) pre-cut at 45° with respect to the load axis (Appendix Figure A.1). Descriptions of the experimental setup, data recording and data treatment can be found in the Appendices A1 - A3, experimental conditions are listed in Table 2.

## 2.2   Strain determination

The thickness of the shear zone at the hit-point is $th_0 = 0.75 \pm 0.03$ mm. During the experiment, $\sim 86 \pm 3$ % of the axial

displacement is accommodated as shear displacement within the shear zone, and $\sim 14 \pm 3$ % is accommodated as plane strain thinning of the shear zone. As in previous experiments, the shear zone thickness decreases linearly with the applied axial displacement (see Marti et al., 2017).

The shear strain is presented as apparent shear strain, $\gamma_a$, and calculated as the sum of the incremental shear displacements divided by the instantaneous shear zone thickness and strain rates are given as apparent shear strain rates, $\dot{\gamma}_a$, (see Marti et al.

2017). $\gamma_a$, is calculated for the full width of the shear zone ignoring any strain localization. Since $\gamma_a$ overestimates the shear strain and cannot be used to determine the strain ellipsoid or other strain related parameters, the procedure described by Fossen and Tikoff (1993); Tikoff (1995) is adopted. Table 2 lists the simple shear ($\gamma$) and pure shear (k) components, as well as the orientation of the instantaneous stretching axes (ISA), the orientation of the finite stretching direction, the kinematic vorticity number and the strain ration given by the ratio of long to short axis of the strain ellipsoid.

## 2.3   Image analysis

After the experiments, samples are immersed in epoxy, cut parallel (in some cases also normal to the shear direction), and prepared to polished thin sections. Polarized light microscope, scanning electron microscope (SEM) and transmission electron microscope (TEM) are used for sample analysis. Grain size and surface fabric are determined as described in Appendix A5.

A special method is developed to study the amphibole coronas that grow on pyroxene porpyhroclasts. Corona thickness is

measured as a function of orientation around the clasts (Figure 1). To this end, phase maps of pyroxene and amphibole are created (as described in Appendix A5). Where amphibole coronas of neighboring pyroxene clasts are in contact, individual pyroxene - amphibole pairs have to be separated manually. Clean phase maps contain segmented pyroxene and amphibole phases, and each pyroxene grain is in contact only with its own amphibole corona. The x-y coordinates of the clast (pyroxene) and the aggregate (pyroxene + corona) outlines are measured and exported using Fiji and a modified version of the *Jazy XY*

*export macro* (by Rüdiger Kilian, available at https://earth.unibas.ch/micro/index.html). Using a MATLAB script (available from the author upon request), the x-y coordinates of clast and aggregate outlines are converted to polar coordinates ($r$-$\theta$), and the corona thickness, *thc($\theta$)*, is determined at each point along the pyroxene clast as the shortest distance between the clast to the aggregate outline (Figure 1c) as a function of $\theta$. The angle runs counter clock wise from the horizontal. This approach yields good results where coronas follow the clast shape, but tends to underestimate corona thickness where the corona becomes very

elongated as e.g. in 'tails' around the clasts. Note that where tails grew extensively long, they were eventually cut, so that the analysis does not include the whole tail length (Figure 1b).





## 3   Results

### 3.1   Mechanical data

For all experiments, the mechanical data plotted as shear stress, $\tau$, vs. apparent shear strain, $\gamma_a$, show a curve with an initial
steep increase of shear stress, reaching a peak value usually after $\sim \gamma_a$ of 0.8 - 1.0 (Figure 2). Peak stress is followed by a slow
decrease, often approaching a quasi-steady state shear stress value from $\gamma_a \approx 4$ onwards. The samples with 0.12 $\mu$l $H_2O$ added
show higher peak stresses and a more rapid shear stress decrease thereafter, compared to samples with 0.2 $\mu$l $H_2O$ added. For
the Maryland Diabase samples (Figure 2a) at Pc $\approx$ 1.0 GPa, the sample with 0.12 $\mu$l $H_2O$ reaches a higher peak stress, but after
an additional $\sim 0.5 \gamma_a$, it drops to a value similar to the samples with 0.2 $\mu$l $H_2O$. Sample strength of Maryland Diabase at 1.0
and 1.5 GPa reach the same peak stress, but the 1.0 GPa experiments weaken more rapidly within the attained strain range.
The synthetic plagioclase-pyroxene mixtures (Figure 2b) show similar peak stress values (460 - 530 MPa) and all but the
An60+En mixture attain similar flow stresses. The synthetic mixtures generally support $\sim$ 60 - 110 MPa higher shear stresses
than the Maryland Diabase samples (compare Figure 2a and b). At peak stress, the synthetic mixtures (samples 503, 518 and
519) reach differential stress values near the Goetze criterion. According to Kohlstedt et al. (1995), the Goetze criterion, $\Delta\sigma$
$\leq$ Pc, is an empirically defined stress range where rocks are expected to deform plastically. However, due to the significant
weakening subsequent to peak stress, many samples which start above the Goetze criterion then fall substantially below it. The
Maryland Diabase samples all stay below the Goetze criterion for all stages of deformation.
Strain rate stepping tests on Maryland Diabase sample material at Pc $\approx$ 1.0 and 1.5 GPa have been performed (Figure 2c).
Stress exponents, $n$, of n = 1.9 and n = 1.4 are obtained for experiments at Pc $\approx$ 1.0 and 1.5 GPa respectively (Figure 3).

### 3.2   Microstructures

In all experiments strain is partitioned into a network of shear bands (Figure 4, Figure 5). Their thickness is variable but the
main shear band strands usually have a thickness on the order of 40 - 150 $\mu$m (e.g. Figure 5d, j) and are characterised by strong
grain size reduction (Figure 5b, e, k).
The following hydration reactions are observed within shear bands and in low strain lenses:


$$Px + Pl + H_2O \longrightarrow Amp + Qz \tag{R1}$$

$$Pl_1 + H_2O \longrightarrow Pl_2 + Zo + Qz + Ky \tag{R2}$$

where $Pl_2$ has a lower anorthite component than $Pl_1$.
In Maryland Diabase samples, both reaction (R1) and (R2) occur pervasively, with reaction (R1) being the more prominent
one. Amphibole grows as reaction coronas on pyroxene clasts and as aggregates, often mixed with quartz, inside shear bands
(Figure 5b, Figure 6a, c). The volume of hydrous reaction products reaches about 15 $-$ 25 % for experiments with durations




of $\sim$ 60 - 70 h (lead run-in and subsequent deformation to $\gamma_a \approx 4$ to 6). Shear bands in Maryland Diabase experiments are broad and subparallel to the shear zone boundaries (Figure 5c), with an angle $\phi = 3°$ between the preferred orientation of

shear bands and shear zone boundaries (see Figure 5 for reference frame). Shear bands are mainly formed by grains with < 1 $\mu$m diameter and frequently show a compositional layering between plagioclase dominated and amphibole dominated layers (Figure 5b, Figure 6a, b). Plagioclase layers are either monomineralic or show mixing with zoisite. In amphibole dominated layers, amphibole is frequently occurring together with quartz. Mixing between amphibole and plagioclase is subordinate.

In synthetic mixtures of An60+En and An60+Di mixtures (Figure 5d - i), (R2) is the dominant hydration reaction. The volume

of hydrous reaction products reaches 1- 9 % for experiments with durations of $\sim$ 66 - 69 h (lead run-in and subsequent deformation). In the An60+En mixture, shear bands are somewhat narrower and more anastomosing. At an angle, $\phi = 9°$, they are also more inclined to the shear zone boundaries compared to the other samples (Figure 5f). Shear bands in both, An60+En and An60+Di mixtures are mainly formed by fine-grained (< 1$\mu$m) plagioclase and zoisite.

The zoisite reaction predicts the formation of a new plagioclase with a lower anorthite component. The fine grain size within

shear bands does not allow for quantitative EDS measurements, but Back-scattered Electron (BSE) images reveal lower Z-contrast (= lower anorthite contents) for plagioclase within shear bands compared to plagioclase porphyroclasts (Figure 5e). Semi-quantitative EDS measurements yield a decrease in anorthite component from $\sim$ An(60) (starting composition) to $\sim$ An(52-55) for grains within shear bands (Figure 7).

In the Ab+En sample, shear bands are broad and sub-parallel to the shear zone boundaries, with $\phi = 6°$ (Figure 5l). Shear

bands are pre-dominantly composed of fine-grained plagioclase (Figure 5k) with sizes < 2 $\mu$m. No difference in composition between plagioclase porphyroclasts and fine-grained plagioclase within shear band was detected. In high-resolution BSE images, a fine-grained phase with a Z-contrast similar to enstatite is observed. Due to the small grain size EDS measurements are extremely challenging but point to a new type of pyroxene with a somewhat higher Si and Na component.

In all experiments (Maryland Diabase and synthetic mixtures), plagioclase shows extensive grain size refinement. Porphyroclasts are replaced by fine-grained plagioclase, nucleating mainly along porphyroclast rims and along straight internal trails, which are thought to represent former fractures (Figure 5k). The newly nucleated grains generally show a lower anorthite component than the plagioclase in the starting material (Figure 7). In experiments on Maryland Diabase, pyroxene grain size reduction is largely caused by the pyroxene-consuming reaction to Amp (R1, Figure 5b, Figure 6). In the synthetic mixtures

however, much of the grain size reduction of pyroxene is caused by fracturing.

The main difference between the microstructures developed at 1.0 and 1.5 GPa Pc (Maryland Diabase experiments) is the increased amount of reaction products at higher Pc (Figure 6). Zoisite and amphibole form more abundantly at 1.5 GPa and amphibole corona surround pyroxene porphyroclasts in early stages of the experiments. Shear bands at Pc $\approx$ 1.0 GPa are

mainly composed of a fine-grained mixture of Pl+Amp+Qz+Zo (in order of abundance), as compared to Amp+Pl+Zo+Qz (again in order of abundance) at 1.5 GPa. Additionally, shear bands are somewhat narrower and more inclined to the shear zone boundaries at the higher Pc (compare Marti et al., 2017).





### 3.3 Amphibole chemistry

For Maryland Diabase experiments at 1.0 GPa, two groups of amphibole are recognized, differing in their Al and Mg per
formula unit (p.f.u.), and in their Na to Al ratio (Figure 8, Table 3, Supplementary Table 1). The amphiboles are classified
as ranging between Tschermakite and Mg-Hornblende. When labelling the amphibole measurements according to their 2D
neighbourhood as observed in the thin section, the Al and Mg contents shows a consistent pattern where high Al - low Mg
amphibole grow in plagioclase dominated areas (Figure 8b - d). The Si and Ca contents thereby show no systematic difference
between the different grain neighbourhoods. Supplementary Table 1 lists amphibole and plagioclase (starting material and
newly nucleated) compositions.

Two groups of amphibole compositions are present, which can be distinguished by their Na per Al content ratio (Figure 8e,
f). The plagioclase of the Maryland Diabase starting material has an anorthite component of $\sim$ An(65-70), with thin rims of
$\sim$ An(52-56) (Figure 7). The core's Na to Al ratio thus is $\sim$ 0.18 - 0.21. Plagioclase is the sole provider for Na and Al in
amphibole as pyroxene in the starting material shows only trace amounts of these elements. Most amphibole measurements
show an Na:Al-ratio of 0.16 - 0.21 (Figure 8f), consistent with reaction (R1) and the consumption of a plagioclase with a
composition of $\sim$ An(65-70). The second type of ampibole, with Na:Al-ratios > 0.25 is comparable to the Na:Al-ratio of the
starting plagioclase rim composition of $\sim$ An(52-56) (resulting in Na:Al-ratios of $\sim$ 0.28 - 0.32) and thus again would be
compatible with the plagioclase-consuming, amphibole forming reaction (R1).

### 3.4 Shear bands

#### 3.4.1 Nanostructure of plagioclase within shear bands

TEM images are presented from shear bands formed within the Ab+En sample 518 (Figure 9) and the Maryland Diabase
sample 414 (Figure 10). For both samples, micrographs are obtained from foils cut normal to the shear zone boundaries and
parallel to the shear direction. Figure 9a shows the interface between an albite porphyroclast and the fine-grained albite matrix
of an adjacent shear band. The albite clast has a high defect density, where intergranular domains develop misorientations to
each other, as seen in the bright field image or from the rotation of diffraction spots (Figure 9b). However, no recovery to
form sub-grain walls is observed. Furthermore, the interface between the clast and the shear band is sharp and no bulges are
observed (Figure 9a). Within the shear band, small pores are seen as pore trails along grain boundaries (Figure 9c, d) oriented
at a small angle to the expected $\sigma_1$.

The shear band formed in the Maryland Diabase sample shows the typical compositional layering between plagioclase domi-
nated layers and amphibole (+Qz) aggregates (Figure 10a). Bright field TEM images reveal largely defect free grains (Figure
10b, d) and grain sizes are similar for amphibole and plagioclase. Grain and phase boundaries are tight and porosity is scarce
(Figure 10c, d). Plagioclase grains are weakly anisotropic in shape (not perfectly equant) with a shape preferred orientation
sub-parallel to the shear zone boundaries (Figure 10b, d; compare Marti et al., 2017).

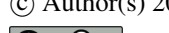


### 3.4.2 Plagioclase grain size distribution within shear bands

2D grain size distributions (GSDs) are determined for plagioclase inside shear bands of the Ab+En experiment 518 and the MD experiment 414 (Figure 11). The distributions in the two samples are similar with somewhat higher frequencies in bins > 1 $\mu$m for albite compared to the labradorites of the Maryland Diabase sample. Due to the small grain sizes and extremely narrow grain boundaries, grains are difficult to identify on SEM images and there is some uncertainty concerning the measured GSDs inherent from the grain segmentation. Nonetheless, grains segmented from TEM and SEM images correlate well for

the highest frequency bins. Measured on TEM and SEM images, the GSDs for sample 518 have modes at 0.51 and 0.36 $\mu$m respectively. In sample 414, the GSDs have modes at 0.23 and 0.30 $\mu$m (Figure 11).

### 3.4.3 Albite crystallographic preferred orientation

Three EBSD maps are collected along one shear band in the Ab+En sample 518 (Figure 12). Orientation data of plagioclase grains with < 2 $\mu$m equivalent diameter are plotted in pole figures for [100], [010] and [001], as well as poles to (010)-

planes. Orientation data of porphyroclasts and their fractured fragments is excluded. Although J-indices indicate only weak crystallographic preferred orientations (CPO) in the different sites, the CPO patterns are similar in all three sites. Poles to (010) planes tend to be oriented near the periphery with a large angle to the trace of the shear band plane. [100] axes show peripheral and central maxima, not far from the trace of the shear band plane. The EBSD map in site2 contains a large plagioclase porphyroclast and the orientation data of this clast is plotted in the pole figures of this site (Figure 12a). The porphyroclast

orientation does not show a systematic relationship with the maxima of the orientations measured from the < 2 $\mu$m shear band grains.

## 3.5 Amphibole coronas

Amphibole grows abundantly in experiments on Maryland Diabase, especially at the higher Pc (e.g. Figure 6c), where pyroxene clasts are surrounded by amphibole coronas already in early stages of the experiment. Pyroxene clast - amphibole rim pairs

from experiments performed at Pc $\approx$ 1.5 GPa were analysed, studying the average amphibole corona thickness as a function of orientation around pyroxene porphyroclasts (Figure 13). Coronas are measured at three different stages: at hydrostatic conditions ($\Delta\sigma \approx 0$), at peak stress ($\Delta\sigma > 0$; $\gamma_a \approx 1$) and after considerable deformation ($\Delta\sigma > 0$; $\gamma_a \approx 4$) corresponding to three evolutionary stages of a typical high strain experiment (Appendix Figure A.1c). Accordingly, three cases are distinguished. The *hydrostatic* case represents the microstructural state at the hit-point after the lead run-in. The *peak stress* case records

the microstructural state at the time the sample has reached its maximum strength (including lead run-in and initial sample loading) and the *deformed* case represents the microstructure evolved after the sample underwent high shear strain (including lead run-in, sample yielding and deformation; for an explanation on the nomenclature used see Appendix Figure A.1c).

At hydrostatic conditions (e.g. during the lead run-in), corona growth is symmetrical around the clasts, with an average thickness of 2.4 to 3.1 $\mu$m (Figure 13a). In the *deformed* case, the average corona thickness shows an overall monoclinic shape.

Assuming that the microstructure after lead run-in (hydrostatic part) is approximately the same as that of the *hydrostatic* case





sample, the corona thickness in the *deformed* sample is reduced by ∼ 0.5 - 2 $\mu$m in directions close to the loading direction. Thickness is reduced in most directions except in the range of 346° - 53° and 186° - 232°, where it is increased. That is, on average, the corona thickness is reduced on clast surfaces facing the loading direction, and increased at high angles to the loading direction. At *peak stress* the average corona thickness in direction of loading is the same as in the *hydrostatic* case (∼

2.7 $\mu$m) but already increased in almost all other directions. Furthermore, despite the 23 h longer duration of the *hydrostatic* run compared to the *peak stress* run (Figure 13c), coronas did not grow to larger thicknesses in the former.

Instantaneous stretching and shortening axes (ISA), finite stretching directions and vorticity numbers (Wk) are calculated for the *peak stress* and the *deformed* case (Figure 13b). The orthorhombic shape of the peak stress corona curve is well described by the ISA, e.g., such that the long side is normal to the shortening ISA. After deformation, the long diameter of the monoclinic

shaped corona curve is oriented between the stretching ISA and the finite stretching direction. The short diameter corresponds to the shortening sector around the pyroxene clast. The direction of the maximum corona thickness is at a higher angle with respect to the shear plane than the finite stretching direction (Figure 13b).

## 4 Discussion

### 4.1 Physics and chemistry of grain size reduction

There is a drastic grain size decrease (down to diameters < 2 $\mu$m) accompanied with the shear band formation (Figure 5, Figures 9-11). Fracturing as an important process of grain size reduction is only observed in pyroxene grains of the synthetic mixtures (e.g. Figure 12b). In these samples, pyroxene only participates in mineral reactions to a minor degree, and the plagioclase hydration reaction (R2) is dominating (except for the Ab+En sample). In contrast, pyroxene grains in experiments on Maryland Diabase show grain size reduction by dissolution during the pyroxene-consuming reaction (R1) to amphibole.

Fracturing only minorly contributes to the grain size reduction of plagioclase. Instead, grain size reduction is primarily caused by mineral reactions and abundant nucleation of new grains. New plagioclase grains have a different composition from that of the original clasts (e.g. Figure 7). The low defect densities, the narrow grain size range, and the lozenge shaped grains of the very fine-grained pure plagioclase aggregates within shear bands (Figures 9-11) are in accordance with a formation by nucleation and limited growth. Of the two initial mineral phases, plagioclase and pyroxene, plagioclase is particularly susceptible to

grain size reduction via reaction and nucleation of new grains. For all new phases, like zoisite and amphibole, it is clear that reaction and nucleation are the mechanisms leading to a small grain size and to phase mixing.

In the special case of the Ab+En experiment, it is difficult to observe an obvious change in chemical composition of the plagioclase and to connect it to grain size reduction in shear bands. Qualitative EDX measurements reveal possible new pyroxene grains with higher Si and Na contents compared to the starting pyroxene. Due to the very small grain size, however, chemical

measurements are challenging. No measurable change in plagioclase composition is detected but in order for the new pyroxene to grow with higher Si and Na contents compared to the starting material, a plagioclase with a higher anorthite component is expected to grow.

Microstructural evidence for grain size reduction by fracturing or dynamic recrystallization (e.g. subgrain rotation or bulging




recrystallization) of plagioclase is not observed (see e.g. Figure 9). In addition, had the recrystallization of plagioclase in

monomineralic domains taken place by dynamic recrystallization, the resulting grain sizes would imply very high stresses. Using the normalized grain size/stress relationship by Derby (1991), the observed plagioclase grain size mode of $\sim 0.4$ $\mu$m (Figure 11) would require differential stresses of 2 to 2.5 GPa. The observed differential stresses are $\sim 500$ MPa in the last stages of this experiment (Figure 2) - they are clearly far too low to produce such a small grain size in equilibrium. As in other samples, grain size reduction of plagioclase in the Ab+En sample is considered to take place by dissolution of original

porphyroclasts and nucleation (i.e. "neo-crystallization") of new grains.

## 4.2    Derivation of the stress exponent

The determined *n*-values are low, with n = 1.4 and 1.9 (Figure 3) and are thus within the range of expected values for diffusion creep (including grain boundary sliding) with theoretical values between 1 (e.g., Ashby and Verrall, 1973; Coble, 1963; Karato, 2008; Kohlstedt and Hansen, 2015; Paterson, 2013), up to 2 (e.g., Gratier et al., 2009, 2013; Paterson, 2013). The stress

exponents determined in this study have to be taken with some caution as deformation of the samples is inhomogeneous. While the shear bands are able to accommodate higher strain rates, the lesser deformed domains between seem to still control the overall bulk stress (Marti et al., 2017). Nevertheless, the low stress exponents strongly suggest an absence of frictional deformation and make dislocation creep unlikely.

## 4.3    Albite crystallographic preferred orientation

Dislocation creep and dynamic recrystallization are not considered to occur in our experiments. The large Burgers vectors and the cation ordering (coupled Al+Ca and Si+Na) in plagioclase are unfavourable for intracrystalline deformation, especially at the comparatively low experimental temperatures of 800 °C.

Dislocation glide and climb have been suggested to be active in plagioclase at both natural and experimental conditions (e.g., Tullis and Yund, 1985; Shaocheng and Mainprice, 1987; Yund and Tullis, 1991; Rybacki and Dresen, 2000; Shigmeatsu and

Tanaka, 2000; Kruse et al., 2001; Lapworth et al., 2002; Stünitz et al., 2003; Ji et al., 2004; Barreiro et al., 2007; Mehl and Hirth, 2008) but usually are not considered to accommodate large amounts of strain. Recrystallization takes place by different mechanisms including neo-crystallization (e.g., Fitz Gerald and Stünitz, 1993; Rosenberg and Stünitz, 2003; Brander et al., 2012; Fukuda and Okudaira, 2013; Mukai et al., 2014) or by growth of fragments formed by fracturing (e.g., Stünitz et al., 2003; Viegas et al., 2016). In fine-grained aggregates, diffusion creep (in the broadest sense) often dissolution-precipitation

creep (DPC), is the main strain accommodating process described for polycrystalline plagioclase aggregates (e.g., Yund and Tullis, 1991; Fitz Gerald and Stünitz, 1993; Jiang et al., 2000; Lapworth et al., 2002; Rosenberg and Stünitz, 2003; Brander et al., 2012; Fukuda and Okudaira, 2013; Mukai et al., 2014; Viegas et al., 2016).

The CPO measured in a shear band of the Ab+En sample 518 is generally weak, but the three independent sites 1 - 3 show very similar CPO patterns (Figure 12). This similarity indicates that these CPOs, although weak, are not random but that there

must be a mechanism leading to this weak but consistent CPO.

There are a number of mechanisms that can lead to a CPO within an aggregate. Examples are dislocation glide (e.g., Schmid





and Boas, 1950) or directed growth (possibly together with rigid body rotation; e.g. Shelley, 1994; Berger and Stünitz, 1996; Rosenberg and Stünitz, 2003; Getsinger and Hirth, 2014; Viegas et al., 2016). Furthermore, a CPO can form due to interfacial energy, e.g., via host-controlled nucleation (e.g., Jiang et al., 2000) or caused by interface-controlled diffusion creep (e.g., Bons and den Brok, 2000; Sundberg and Cooper, 2008). Similar CPOs as the ones shown here have been found in experimentally deformed anorthite and basalt (e.g., Ji et al., 2004; Barreiro et al., 2007), and in naturally deformed basaltic and peridotitic rocks (e.g., Mehl and Hirth, 2008; Viegas et al., 2016; Xie et al., 2003; Drury et al., 2011). Ji et al. (2004) and Mehl and Hirth (2008) interpret the CPOs to be due to dislocation creep in monomineralic plagioclase layers. Viegas et al. (2016) observe no evidence for dislocation creep and suggest that the CPOs may be the result of directed growth and rigid body rotation of grains (with a crystallographically controlled grain shape) during diffusion accommodated grain boundary sliding. In our case, we observe no microstructural evidence for dislocation creep and suggest that the observed CPOs did not form as a consequence of intracrystalline plasticity (see also section 4.5).

### 4.4 Evidence for dissolution-precipitation creep of amphibole

During the experiments, amphibole grows as a reaction product and pyroxene porphyroclasts are replaced at their rims by amphibole growth coronas. During all stages of an experiment, amphibole is observed to grow stably. However, the amphibole coronas, which grow symmetrically in thickness during the hydrostatic 'lead run-in', become partly dissolved in high stress sites while simultaneously growing thicker in low stress sites during deformation (Figure 13a). These results indicate grain scale DPC of amphibole. DPC is a form of diffusion creep in the presence of an aqueous fluid. It is frequently observed for amphiboles in naturally deformed rocks (e.g., Berger and Stünitz, 1996; Imon et al., 2002; Marsh et al., 2009; Stokes et al., 2012), but has only rarely been reproduced in experimental studies (Rutter et al., 1985; Getsinger and Hirth, 2014).

The direction of the shortening ISA (equal to the direction of the instantaneous maximum principal stress, $\sigma_1$) lies within the range of directions of minimum average corona thickness. A correlation between the shortening ISA and the minimum in average corona thickness is consistent with the interpretation of DPC, where material is preferentially dissolved along high stress sites (and re-precipitated along low stress sites). The geometry of deformation by diffusion creep is irrotational (Karato, 2008), that is, as the principal stress axes are normal to each other, the resulting (grain shape) fabric due to diffusion creep should be orthorhombic. At the *peak stress*, where shear strain is still small, the rim thickness shows an overall ∼ orthorhombic shape (see corona curve in Figure 13b), well aligned with the ISAs. As the ISAs should indicate the minimum and maximum principle stress directions, the fact that the corona curve follows the ISAs correlates well with the interpretation that DPC determines the amphibole corona thickness evolution. The monoclinic shape of the corona curve in the *deformed* case can be explained by a superposition of co-axial geometry of deformation by diffusion creep, and rigid body rotation induced by the rotational component of simple shear.

Clasts with coronas are predominantly found in low strain lenses, where shear strains are lower than in shear bands. No microstructure can be found to indicate that the reduction in average corona thickness at the compressional sites of the clasts is due to 'shearing-off' of amphibole from compressional to extensional sites by some sort of granular flow.





### 4.5 Dissolution-precipitation creep and grain boundary sliding

From the mechanical data (including stress exponents), the determined grain sizes, from the nucleation of new grains, and from amphibole growth fabrics discussed above, it is concluded that in all samples, the dominant deformation mechanism cannot be frictional or crystal plastic (dislocation creep). Instead, the dominant deformation mechanisms for amphibole and plagioclase are inferred to be DPC (accompanied and/or accommodated by mineral reactions) and grain boundary sliding. Pyroxene is less involved in accommodating strain but plays an important part by being involved in mineral reactions and thereby aiding grain size reduction by nucleation of new grains.

Mineral reactions change the initial phase assemblage of Px+Pl to mostly Pl+Px+Zo in the synthetic mixtures, and Pl+Px+Amp +Qz+Zo in Maryland Diabase samples. Disregarding the differences in amount and type of mineral reactions, strain is always localized into a network of shear bands characterized by intense grain size reduction and phase mixing (to a lesser extent in the Ab+En sample; Figure 5).

The small size of grains in shear bands (Figure 11) clearly favours a grain size sensitive deformation mechanism such as DPC and grain boundary sliding. This interpretation is also supported by the strain free interior and grain morphology of the small grains in shear bands (Figure 9 and 10). The activity of solution-mass transport processes is clearly indicated by the vast extent of mineral reactions, which necessitate the movement of chemical components over several 10's of $\mu$m at least. The morphology of pores presented in Figure 9 is further supporting evidence for DPC interpretation; the pore trail in Figure 9d is interpreted to have formed by precipitation of plagioclase and entrapment of residual fluid along grain boundaries with a trace sub-parallel to the estimated $\sigma_1$ direction (expected opening direction of dilatancy normal to $\sigma_1$).

Concerning the albite CPO measured in the Ab+En sample, it is argued that the weak CPO is not due to dislocation creep, as discussed earlier. Rather, it is suggested that the CPO formed during grain size sensitive creep in the predominantly monomineralic albite layers. In the absence of further investigations it may be speculated that the CPO formation could be due to directed and anisotropic growth of the albite during DPC and grain boundary sliding of grains with a crystallographically controlled shape.

### 4.6 Importance of dissolution-precipitation creep in natural rocks

Grain size reduction is energetically unfavourable as it increases the total grain surface area. DPC does not necessarily require the formation of new grains, instead, precipitation could take place as overgrowth rims on existing grains. The intense grain size reduction observed in shear bands within our experiments is most probably caused by high nucleation rates. High nucleation rates are typically attained by a large overstepping of a reaction boundary (e.g., Rubie, 1998; Putnis, 1992), introducing a high driving potential ($\Delta$Gt). The start of our experiments represents such an instance of large overstepping of reaction boundaries. The starting materials + $H_2O$ are not in equilibrium at the experimental Pc-T conditions. As the pressurization and heating procedure required to attain the experimental Pc-T conditions takes place within 5 – 8 h, the sample material is brought rapidly to a metastable state. Although this rapid change in P-T conditions is unique to experiments, there is widespread evidence from observations of natural rocks that similarly, metastable mineral assemblages can be sustained even at high-grade conditions





and to large overstepping of reaction boundaries, when rocks are dry (e.g., Rubie, 1986; Austrheim, 1987; Wayte et al., 1989;
Krabbendam et al., 2000; Austrheim, 2013; Jamtveit et al., 2016). Only where fluid infiltrates, mineral reactions are enabled
and equilibration can be attained.

TEM analyses revealed high defect densities in the albite porphyroclasts (Figure 9a, b). It has been proposed that a high
intragranular defect density results in an increased rate of grain dissolution, speeding up the reaction and/or deformation rate
(e.g., Wintsch, 1985; Schott et al., 1989; Stünitz, 1998). However, the observation of, e.g., abundant nucleation along former
fractures (Figure 5k) cannot simply be accredited to enhanced reaction rates due to locally increased strain energy (in the sense
of high defect densities locally introduced by fracturing according to, e.g., Fitz Gerald et al., 1991; Fitz Gerald and Stünitz,
1993; Stünitz et al., 2003; Trepmann et al., 2007). Fracturing is always accompanied by dilatancy. Fluid infiltrates the fractures,
leading to higher solution-mass transport rates (e.g., Fitz Gerald and Stünitz, 1993; Precigout and Stünitz, 2016). Thus higher
reaction rates can be expected. The significance of strain energy as a possible rate-enhancing contributor to reaction and
nucleation is thus difficult to separate from the effects of enhanced fluid flow.
With the dominant deformation mechanisms, DPC and grain boundary sliding, the samples presented here deform viscously.
At the experimentally induced conditions of T = 800 °C and at strain rates of $\sim 3 \cdot 10^{-5}$ s$^{-1}$, deformation takes place at the lower
end of the viscous field, close to the brittle-viscous transition of the studied material (e.g., Marti et al., 2017). However, it is
probable that the observed deformation mechanisms are also active at higher temperatures in these types of rocks. The principal
constituent phases of mafic rocks (plagioclase + pyroxene + amphibole) all have high strengths in terms of intracrystalline
plasticity, even at high temperatures. Where water is absent and DCP suppressed, the build-up of high stresses can be expected
in such rocks, at least transiently (e.g., Okudaira et al., 2015). If local stresses are high enough to induce cracking, dilatancy
and fluid infiltration may be facilitated. Once reactions start to operate (metastability of mafic rocks is very common because
the mineral compositions are very variable and critically dependent on P, T, fluid-composition), a switch to DPC is likely to
occur. The resulting deformation takes place with low stress exponents. It is clearly seen in our experiments that DPC and grain
boundary sliding, as grain size sensitive deformation mechanisms, are strongly supported by the extensive mineral reactions
causing grain size reduction by heterogeneous nucleation.

### 4.7 Continued operation of deformation mechanisms at higher strain

In the case of the experimental set-up described here, where the starting mineral assemblages are not in equilibrium at the
imposed P-T-fluid conditions, the chemical driving potential for attaining a lower energy assemblage partially controls the
reaction rate. In principle, when the stable assemblage is reached, i.e., when the reaction has gone to completion, the driving
potential for dissolution of phases is reduced, and it may be expected that the deformation rate is reduced, too. Interestingly,
the measured amphibole compositions show variations, which are most probably influenced by the local mineral composition
of the other phases in the neighbourhood (Figure 8). Shear offset and neighbour switching during deformation by diffusion
creep and grain boundary sliding will continuously change the local neighbourhood of grains. As the neighbourhood of a
given grain changes, so will new surfaces be exposed to the grain boundary fluid and become involved in reactions, potentially
providing a relatively constant chemical driving force for reaction and nucleation, if local disequilibrium prevails. Thus, it may



## 5   Summary and Conclusions

Viscous deformation in experiments with mafic compositions at temperatures of 800 °C and confining pressures of 1.0 and 1.5 GPa at strain rates of $\sim 10^{-5}$ s$^{-1}$ is dominantly achieved by dissolution-precipitation creep (DPC) and grain boundary sliding, accompanied by syn-deformational mineral reactions. No evidence for frictional deformation or significant contributions of dislocation glide or creep to the accommodation of strain can be found for any of the mineral phases. Strain is frequently localised into shear bands, which consist of fine-grained mixtures of neo-crystallized plagioclase and the syn-kinematic reaction products amphibole, quartz and zoisite, none of which are present in the starting material.

Intense grain size reduction is produced by high nucleation rates, probably caused by a large overstepping of reaction boundaries. Both, deformation and nucleation are localised in shear bands, implying a positive feedback between the two mechanisms.

Amphibole is seen to accommodate displacement via dissolution-precipitation creep, as interpreted from the evolution and distribution of amphibole coronas on pyroxene porphyroclasts. A weak but consistent crystallographic preferred orientation (CPO) of albite is formed in shear bands during deformation by DPC and grain boundary sliding.

The deformation in the samples takes place under conditions of pronounced weakening in all cases. The weakening is induced by strain localization into shear bands that show strong grain size reduction. Grain reduction, in turn, is due to nucleation of new phases, demonstrating the direct relationship between mineral reaction, grain size refinement, as well as the operation of DPC and grain boundary sliding (as grain size sensitive mechanisms), resulting in viscous deformation with low stress exponents.

*Code and data availability.*   The MATLAB code for analysing amphibole reaction corona thicknesses is available from the author upon request (sina.marti@ed.ac.uk). Mechanical data and chemical analyses are available from the author upon request.

## Appendix A:  Methods

### A1   Sample preparation

Maryland Diabase rock powder is fabricated by crushing Maryland Diabase pieces with a hand-press and subsequently with an alumina hand-mortar. The resulting powder is dry-sieved to extract a grain size fraction $\leq 125$ $\mu$m. The plagioclase in Maryland Diabase shows a relatively homogeneous composition ($\sim$ An65-70) except for a thin rim with lower anorthite component ($\sim$ An50-55). The core to rim area ratio is   83 : 17 ($\pm$ 3). Some of the clinopyroxene grains show a Mg-enriched core and clinopyroxene grains generally show orthopyroxene exsolution lamellae.

The diopside and enstatite material is a mineral powder provided by Jacques Precigout (University d'Orléans) and Holger




Stünitz (University of Tromso, University d'Orléans) with grain sizes of 40 - 125 $\mu$m for Cranberry Lake diopside, 40 - 180 $\mu$m for Damaping enstatite. Damaping enstatite and diopside are derived from a peridotite xenolith, the Cranberry Lake diopside from a calc-silicate rock. The albite material is extracted from an albite-quartz vein formed along a joint from the Alpe Rischuna area, Switzerland. Sonora labradorite are labradorite megacrysts formed in basaltic deposits from the Pinacate

volcanic field, Sonora, Mexico. From Sonora labradorite, Alpe Rischuna albite and Damaping diopside a powder (grain size fraction $\leq$ 125 $\mu$m) is produced in the same manner as described for the Maryland Diabase powder. As the Sonora labradorite material shows some accessory calcite, the powder is cleaned with $HCl_{aq}$ (10%). Subsequently, the powder is placed in a funnel with a grade 602 h qualitative filter paper with a pore size of 2 $\mu$m and rinsed thoroughly with distilled water. The powder retained by the filter is then dried in an oven at $\sim$ 110 °C. After this treatment, no calcite is detected in the material.

Synthetic plagioclase-pyroxene powders are mixed with a phase distribution of $\sim$ 57 vol% plagioclase to 43 vol% pyroxene. To produce the synthetic mixtures, the powders are placed in a 5 ml glass beaker with acetone and mixed using an ultrasonic stirrer (procedure of de Ronde et al., 2004). When most of the acetone is evaporated, the slurry is dried in an oven at 110 °C. This procedure prevents grain size and density sorting of the minerals.

The sample is then placed in a platinum jacket (0.15 mm wall thickness) with a 0.025 mm nickel foil insert. The Pt-jacket is

weld-sealed with a Lampert welding apparatus while the sample is encased in a cooled brass piece (T $\sim$ 4 °C) to minimize sample heating and resulting potential water loss.

Solid sodium chloride (NaCl) is used as confining medium (Appendix Figure A.1a). K-Type thermocouples (with metal tubing) positioned next to the sample are used for most experiments. Only for long-duration experiments of more than 6 days, S-type thermocouples (with mullite tubing) are used, as the mullite tubing is more durable in the corrosive environment of the heated

salt.

## A2   Experimental procedure

Confining pressure, axial load, and displacement are constantly recorded. The force on the load-piston is measured using an external load cell, whereas the displacement of the load-piston is measured with either a direct current displacement transducer (resolution $\sim$ 1 $\mu$m) (Rig 1) or a noiseless digital linear-transformation measurement system (resolution = 0.1 $\mu$m) (Rig 2). Pc

is measured via the oil pressure in the hydraulic pumping system and T is monitored and regulated to within $\pm$ 1 °C via an Eurotherm controller.

To bring the sample to the desired Pc-T conditions during pressurisation, the independently movable Pc- and load-pistons are alternately advanced (thereby raising the pressure), accompanied by stepwise increases of temperature until the desired Pc-T conditions are reached (Appendix Figure A.1b). The pressurization procedure usually takes 5 – 8 h. During the actual deforma-

tion experiment, only the load-piston is advanced. The experimental setup necessitates that each deformation experiment starts with a so-called 'lead run-in', where the load-piston is advanced through a thin ($\sim$ 1.5 mm) top lead layer. During this stage, the sample is expected to experience $\sim$ isotropic pressure. During the initial lead run-in, the sample is under approximately hydrostatic conditions for several hours (usually between 24 - 30 h). Once the hit point is reached, sample loading initiates. At



the end of an experiment, the sample is quenched to 200 °C within 2 min while simultaneously removing the differential load.
Subsequently Pc, load and T are slowly and simultaneously reduced to room conditions within about 5 h.

## A3   Evaluation of mechanical data

Axial displacement is corrected for apparatus stiffness. $\sigma_3$ is assumed to be equal to the confining pressure, Pc (Eq. (A1)). With increasing advancement of the load-piston, the pressure inside the vessel increases and Pc is corrected for this (see e.g., Richter et al., 2016). The differential stress, $\Delta\sigma$, acting on the sample is derived from the difference between the axial load (F) with
reference to the load at the hit point (F0) (Eq. (A2)), and the cross-sectional area of the forcing block (31.47 mm$^2$)

$$\sigma_3 = Pc \tag{A1}$$

$$\Delta\sigma = (F - F_0)/31.47mm^2 \tag{A2}$$

$\Delta\sigma$ in the sample is corrected for the decreasing overlap of the forcing blocks (see procedure described in Marti et al., 2017). The shear and normal stresses, $\tau$ and $\sigma_n$ supported by the sample inclined at 45° are obtained by Mohr circle construction from
$\Delta\sigma$. The effective pore fluid pressures in our experiments is assumed to be negligible, i.e. is taken as zero and the deviation of $\sigma_1$ by 3-6° from 45° is neglected as well.

Stress exponent, *n*, is calculated for a relationship

$$\tau = \dot{\gamma}_a^{1/n} \tag{A3}$$

between shear stress, $\tau$, and strain rate, $\dot{\gamma}_a$, (Eq. (A3)) using data of displacement-rate stepping and single-displacement rate
experiments.

Over the years and between the different laboratories, different data treatment routines have been developed and are in use. The variations in calculated stresses using the different methods can influence determined stress exponents (*n*) from the data, causing variations on the order of 16 – 27 % on determined n values (see Marti et al., 2017)

## A4   Strain and flow descriptors

The ISA and the finite stretching direction can be derived because the initial thickness of the shear zone is known (with an error of ± 0.03 mm) and the end-thickness can be measured after the experiment from the thin section. This uncertainty associated with the initial thickness leads to some uncertainty concerning the amount of thinning, and therefore also to a range of values for the ISA, the finite stretching direction and Wk.

## A5   Microstructure analysis

*- Data acquisition*
SEM analyses are performed either with the Zeiss Merlin SEM at Tromsø University, or with a Philips XL30 ESEM at the



Nano Imaging Lab of the Swiss Nanoscience Institute (SNI) at Basel University. Chemical analyses are performed using standardless energy dispersive X-ray Spectrometry (EDS), at 15 keV acceleration voltage with a ZAF matrix correction for spectra quantification.

(Scanning) transmission electron microscopy ((S)TEM) analyses are performed at Utrecht University using a FEI Talos 200FX equipped with a high-sensitivity Super-EDX system. TEM images are recorded in bright field (BF) mode, whereas STEM images in high angular annular dark field (HAADF) mode. BF-TEM images are highly sensitive on crystallographic orientation, whereas contrasts in HAADF-STEM images are sensitive to average atomic number (Z-contrast) of the material. Focussed ion beam (FIB) foils for TEM investigations are prepared using an FEI Helios NanoLab G3. The FIB foils are cut parallel to
the shear direction and normal to the shear plane. Both, the FEI Talos 200FX and the Nanolab 3G are located at the Electron microscopy square at Utrecht University.

*- EBSD analysis*

For Electron backscatter diffraction (EBSD) measurements, thin sections were polished with colloidal silica suspension and
subsequently coated with a thin layer of carbon. Samples were analysed in the Zeiss Merlin SEM at the University of Tromso, with a Nordlys nano camera in high vacuum at 15 keV acceleration voltage and probe currents of $\sim$ 15 nA. Data is acquired with the Oxford AZtec software and processed with Channel 5 and the MATLAB toolbox MTEX (available at https://mtex-toolbox.github.io; Bachmann et al., 2010). Pole figures are plotted using MTEX. For pole figure contouring, the de la Vallée Poussin kernel with a half-width of 9.4° and a bandwidth of 30 is used. Pole figure J-index (PFJ; e.g., Bunge, 1982; Mainprice
and Silver, 1993) are given as a measure of texture strength. The index has a value of 1 for a random distribution and is infinite for a single orientation.

*- Image analysis*

Phase segmentation: Phase segmentation is performed on BSE SEM images, where individual minerals are identified by their Z-
contrast. Using the software Fiji (link at: http://fiji.sc/Fiji) and the plugin Statistical Region Merger, automatic pre-segmentation was achieved. The automatic segmentation was then manually inspected and corrected where necessary.

Grain maps: Grain maps were produced by manually tracing (supervised segmentation) grains from SEM or TEM images. Grains were analyzed for their 2D area using Fiji, and for each grain, the diameter of the area equivalent circle, $d_{equ}$, is
calculated.

$$d_{equ} = 2 \cdot \sqrt{(A/\pi)} \tag{A4}$$

where A is the area of the digitized shape. Grain size distributions are presented as histograms of the 2D number weighted distribution of equivalent diameters.

Shear band orientation: Shear bands are traced and digitized on BSE SEM images. The x-y-coordinates of the outlines of the
digitized structures are smoothed to remove pixel artefacts, and exported using Fiji. The smoothed x-y-coordinates are used as



input for the SURFOR program (Panozzo Heilbronner, 1984; Heilbronner and Barrett, 2014), which determines the orientation distribution function (ODF) of the boundary segments. The ODFs are presented as rose diagrams.

*Acknowledgements.*  We thank the team of the centre of nano imaging (SNI) at Basel University and Tom Eilertsen at Tromso University for help and assistance with the electron microscopy. Terry Tullis is thanked for providing the Maryland Diabase material and the Cranberry

Lake Diopside was kindly provided by Jacques Précigout. Willy Tschudin is thanked for excellent thin section preparation. We gratefully acknowledge the funding provided by the Swiss National Foundation grant NF 200020_144448 and financial support from the Freiwillige Akademische Gesellschaft, Basel, during the last stages of finishing this manuscript.



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



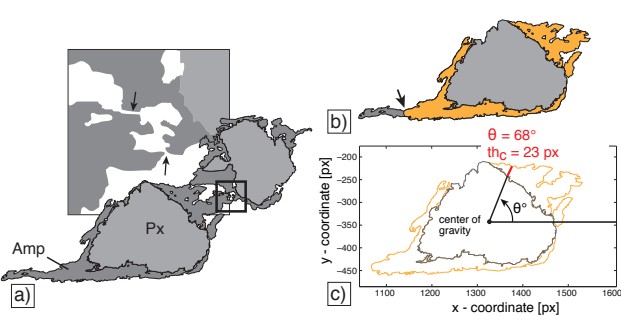

**Figure 1.** Analysis of amphibole corona thickness. a) Digital phase map of segmented Pyroxene (Px) clasts and associated amphibole (Amp) coronas. Where adjacent coronas are in contact, they are separated manually (close-up, black arrows). b) Long 'tails' of Amp growing in low stress sites around clasts are eventually 'cut' (black arrow) if they are extending too far away from the clast. c) Corona thickness, thc($\theta$), is determined from the polar coordinates of the aggregate and clast outline as a function of the angle $\theta$ ($0° < \theta < 360°$), with $\theta$ running CCL from the horizontal.

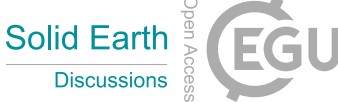



**Table 1.** Mineral composition. Representative mineral measurements as normalised oxide wt% and as calculated stoichiometric mineral composition for the different starting materials. All Fe is taken as $Fe^{2+}$ due to the reducing environment in the sample assembly.

| wt.-% | Alpe Rischuna Albite | Sonora Labradorite | Maryland Diabase Plagioclase core | rim | wt.-% | Cranberry lake Diopside | Damaping Diopside | Damaping Enstatite | Maryland Diabase Cpx | Maryland Diabase Opx |
|---|---|---|---|---|---|---|---|---|---|---|
| $SiO_2$ | 67.87 | 53.66 | 51.86 | 55.67 | $SiO_2$ | 57.18 | 54.39 | 55.98 | 51.58 | 52.61 |
| $Al_2O_3$ | 20.14 | 30.37 | 29.92 | 27.72 | $Al_2O_3$ | 0.85 | 6.37 | 4.01 | 1.77 | 0.75 |
| CaO | 0.21 | 11.09 | 13.39 | 10.57 | CaO | 22.54 | 17.93 | 0.80 | 14.71 | 1.44 |
| $Na_2O$ | 11.71 | 3.90 | 3.63 | 5.11 | $Na_2O$ | 0.39 | 1.71 | 0.27 | 0.28 | 0.00 |
| $K_2O$ | 0.10 | 0.39 | 0.26 | 0.37 | $K_2O$ | 0.00 | 0.00 | 0.00 | 0.00 | 0.00 |
| MgO | 0.00 | 0.00 | 0.00 | 0.00 | MgO | 17.93 | 16.13 | 33.28 | 14.03 | 19.36 |
| $TiO_2$ | 0.00 | 0.00 | 0.00 | 0.00 | $TiO_2$ | 0.00 | 0.34 | 0.00 | 0.76 | 0.28 |
| FeO | 0.00 | 0.59 | 0.94 | 0.55 | FeO | 1.11 | 2.37 | 5.13 | 16.40 | 25.55 |
| MnO | 0.00 | 0.00 | 0.00 | 0.00 | MnO | 0.00 | 0.00 | 0.00 | 0.48 | 0.00 |
| $Cr_2O_3$ | 0.00 | 0.00 | 0.00 | 0.00 | $Cr_2O_3$ | 0.00 | 0.76 | 0.53 | 0.00 | 0.00 |
| Total: | 100.00 | 100.00 | 100.00 | 99.99 | Total: | 100.00 | 100.00 | 100.00 | 100.01 | 99.99 |
| **Atoms per 8 oxygen** | | | | | **Atoms per 6 oxygen** | | | | | |
| Si | 2.97 | 2.42 | 2.36 | 2.51 | Si | 2.04 | 1.94 | 1.92 | 1.95 | 1.99 |
| Al | 1.04 | 1.61 | 1.61 | 1.47 | Al | 0.04 | 0.27 | 0.16 | 0.08 | 0.03 |
| Ca | 0.01 | 0.54 | 0.65 | 0.51 | Ca | 0.86 | 0.68 | 0.03 | 0.60 | 0.06 |
| Na | 0.99 | 0.34 | 0.32 | 0.45 | Na | 0.03 | 0.12 | 0.02 | 0.02 | 0.00 |
| K | 0.01 | 0.02 | 0.02 | 0.02 | K | 0.00 | 0.00 | 0.00 | 0.00 | 0.00 |
| Mg | 0.00 | 0.00 | 0.00 | 0.00 | Mg | 0.95 | 0.86 | 1.70 | 0.79 | 1.09 |
| Ti | 0.00 | 0.00 | 0.00 | 0.00 | Ti | 0.00 | 0.01 | 0.00 | 0.02 | 0.01 |
| Fe | 0.00 | 0.02 | 0.04 | 0.02 | Fe | 0.03 | 0.07 | 0.15 | 0.52 | 0.81 |
| **Mn** | 0.00 | 0.00 | 0.00 | 0.00 | **Mn** | 0.00 | 0.00 | 0.00 | 0.02 | 0.00 |
| **Cr** | 0.00 | 0.00 | 0.00 | 0.00 | **Cr** | 0.00 | 0.02 | 0.01 | 0.00 | 0.00 |
| Total | 5.01 | 4.96 | 5.00 | 4.97 | Total | 3.95 | 3.97 | 4.00 | 4.00 | 3.99 |
| An | 0.98 | 0.60 | 0.66 | 0.52 | En | 0.52 | 0.53 | 0.91 | 0.42 | 0.56 |
| Ab | 98.47 | 0.38 | 0.32 | 0.46 | Fe | 0.02 | 0.04 | 0.08 | 0.27 | 0.41 |
| Or | 0.55 | 0.02 | 0.02 | 0.02 | Wo | 0.47 | 0.42 | 0.02 | 0.31 | 0.03 |





**Table 2.** Experimental conditions.

| Exp. nr. | Material | Pc [GPa] | peak $\tau$ [MPa] | $\tau$ at end [MPa] | mean $\dot{\gamma}_a$ [s$^{-1}$] | μl H$_2$O added | $\gamma_a$ | th0 [mm] | thF [mm] | ds [mm] | $\gamma$ | k | wk | shortening ISA [°] | $\Psi$ [°] | R |
|---|---|---|---|---|---|---|---|---|---|---|---|---|---|---|---|---|
| 414 | MD | 0.97 | 407 | 192 | 2.1e-5 [1] | 0.20 | 5.12 | 0.75 | 0.50 | 2.56 | 3.32 | 1.50 | 0.972 | 38.1 | 9.3 | 14.3 |
| 449 | MD | 1.50 | 479 | 337 | 2.3e-5 [1] | 0.20 | 4.51 | 0.75 | 0.61 | 2.75 | 3.64 | 1.23 | 0.994 | 41.8 | 11.3 | 15.6 |
| 468* | MD | 1.07 | 348 | | 1.2e-5 [3] | 0.20 | 0.70 | 0.75 | 0.69 | 0.48 | 0.64 | 1.09 | 0.968 | 37.7 | 28.8 | 1.9 |
| 470* | MD | 1.50 | 446 | | 1.3e-5 [3] | 0.20 | 0.86 | 0.75 | 0.69 | 0.65 | 0.86 | 1.09 | 0.982 | 39.5 | 27.9 | 2.3 |
| 484** | MD | 1.02 | 371 | 316 / 233 | 1.9e-5 [2] / 9.5e-6 [2] | 0.20 | 1.97 | 0.75 | 0.56 | 1.33 | | | | | | |
| 489 | MD | 1.05 | 428 | 286 | 1.9e-5 [1] | 0.20 | 3.04 | 0.75 | 0.63 | 1.91 | 2.54 | 1.19 | 0.991 | 41.1 | 15.3 | 8.5 |
| 490** | MD | 1.00 | 350 | 130 | 2.4e-5 [2] / 2.2e-5 [2] / 2.5e-6 [2] | 0.20 | 5.37 | 0.75 | | | | | | | | |
| 491** | MD | 1.52 | 388 | 82 | 1.7e-5 [2] / 8.0e-6 [2] / 2.3e-6 [2] | 0.20 | 4.97 | 0.75 | | | | | | | | |
| 492 | MD | 1.01 | 468 | 197 | 2.6e-5 [1] | 0.20 | 8.95 | 0.75 | 0.44 | 3.94 | 5.01 | 1.70 | 0.978 | 39.0 | 5.8 | 30.8 |
| 502** | MD | 1.52 | 391 | 33 | 1.8e-5 [2] / 8.1e-6 [2] / 1.7e-6 [2] | 0.20 | 3.90 | 0.75 | | | | | | | | |
| 503 | An60+En | 1.03 | 530 | 367 | 2.6e-5 [1] | 0.12 | 6.55 | 0.75 | 0.58 | 2.26 | 2.98 | 1.29 | 0.985 | 40.1 | 12.3 | 11.3 |
| 505 | An60+Di (CrLk) | 1.01 | 460 | 253 | 2.5e-5 [1] | 0.12 | 6.17 | 0.75 | 0.55 | 3.60 | 4.73 | 1.36 | 0.992 | 41.3 | 8.1 | 25.4 |
| 507 | MD | 1.04 | 479 | 191 | 2.4e-5 [1] | 0.12 | 5.82 | 0.75 | 0.55 | 3.39 | 4.45 | 1.36 | 0.990 | 41.0 | 8.5 | 22.8 |
| 518 | Ab+En | 1.02 | 511 | 263 | 2.4e-5 [1] | 0.12 | 5.83 | 0.75 | 0.49 | 2.85 | 3.69 | 1.53 | 0.974 | 38.5 | 8.4 | 17.2 |
| 519 | En60+Di (D) | 1.02 | 517 | 289 | 2.2e-5 [1] | 0.12 | 5.60 | 0.75 | 0.54 | 3.15 | 4.12 | 1.39 | 0.988 | 40.5 | 8.8 | 20.0 |
| | | | | | | | | | 0.54 | 3.02 | 3.96 | 1.39 | 0.987 | 40.3 | 9.1 | 18.6 |

MD=Maryland Diabase, Slab=Sonora Labradorite, En=Damaping Enstatite, Di (CrLk)=Cranberry lake Diopside, Di(D)=Damaping Diopside, Ab=Alpe Rischuna Albite. * experiment terminated at peak-stress; ** displacement-rate stepping test. $\Psi$ = finite stretching direction. $\Psi$ = finite stretching test. $\Psi$ = finite strain ellipse major vs minor axis.





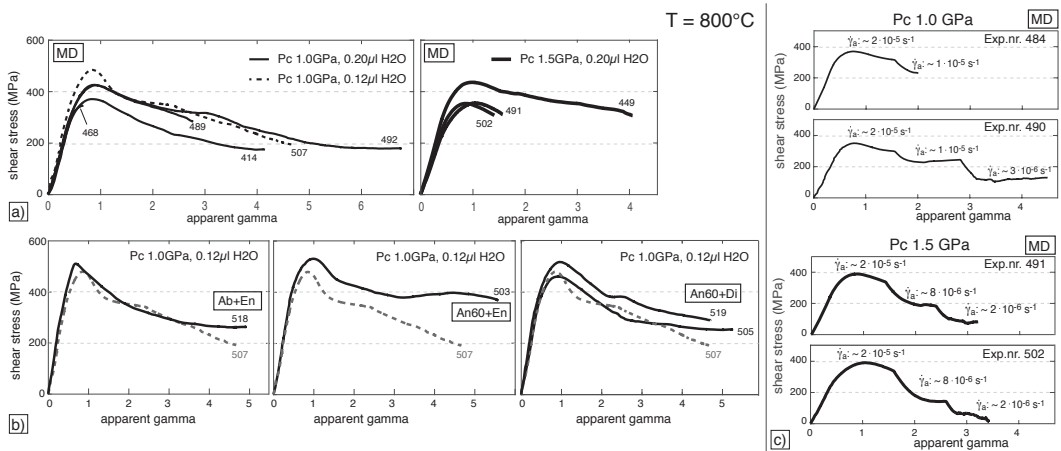

**Figure 2.** Mechanical data. Shear stress, $\tau$ (MPa), versus apparent shear strain, $\gamma_a$. Stippled line = Experiment 507 (MD) for reference. a) Maryland Diabase (MD) experiments for different confining pressures, Pc (GPa), and water contents. b) Experiments using different Pl-Px mixtures and constant Pc and water content. c) Displacement-rate stepping tests on MD sample material for experiments performed at different confining pressures.





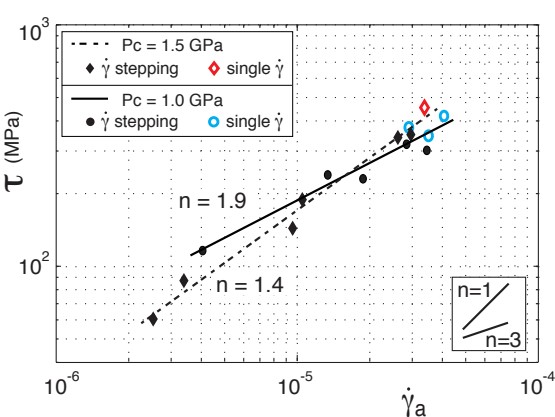

**Figure 3.** Determination of stress exponents. Shear stress, $\tau$ (MPa), versus apparent shear strain rate, $\dot{\gamma}_a$. Two stress exponents, n, are obtained using constant strain rate data and strain rate stepping experiments. For experiments at confining pressures, Pc = 1.0 GPa, n = 1.9, for Pc = 1.5 GPa, n = 1.4. Data for Pc = 1.0 GPa from Marti et al. (2017).




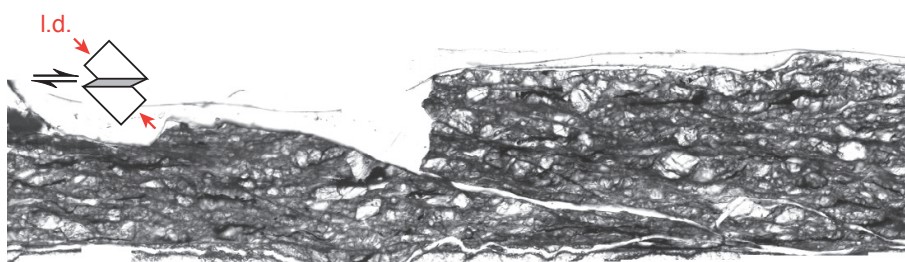

**Figure 4.** Shear zone overview. Micrograph of sample 492, plane polarized light. Strain localizes into a network of shear bands, anastomosing around low strain lenses identifiable by the large porphyroclasts. Sketch in upper left shows orientation of the micrograph with respect to the loading direction (l.d.) of the sample setup (Appendix Figure A.1).



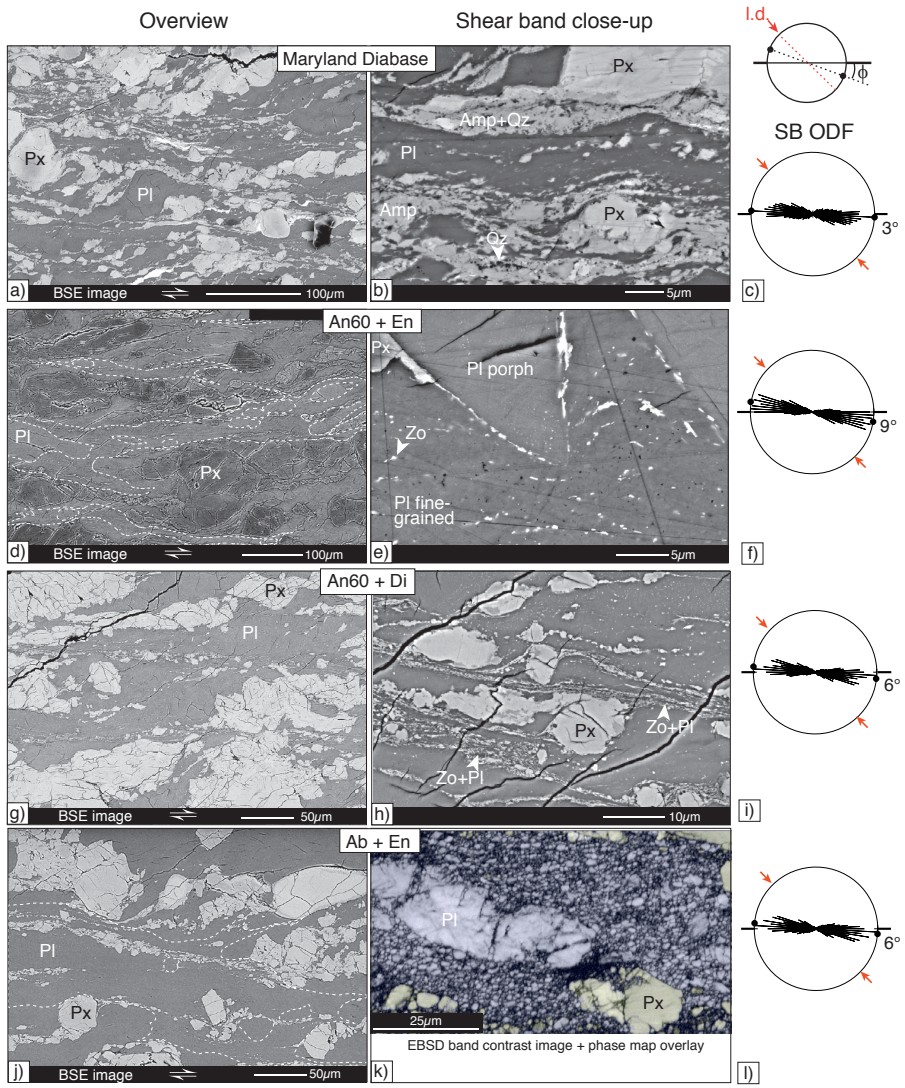

**Figure 5.** Microstructures of experiments at Pc ≈ 1.0 GPa. a) - c) Maryland Diabase sample material. b) Shear bands are fine-grained and often polymineralic, with the main constituents Pl, Amp and Qz. d) - f) An60+En sample material. Due to the low iron content, pyroxene appears darker than the plagioclase in BSE SEM images. In d) and j), shear bands are traced with white dotted lines. e) Fine-grained Pl + Zo in a shear band next to a Pl porphyroclast. g) - i) An60+Di sample material. j) - l) Ab+En sample material. h) EBSD band contrast image with transparent phase map overlay. Plagioclase appears blueish, pyroxene yellowish. Rose diagrams represent the orientation of the shear bands, black dots indicate preferred trend of traces. Red arrows indicate direction of loading direction. Angle, $\phi$, between the shear zone boundary (or forcing block) and the preferred trend is indicated.





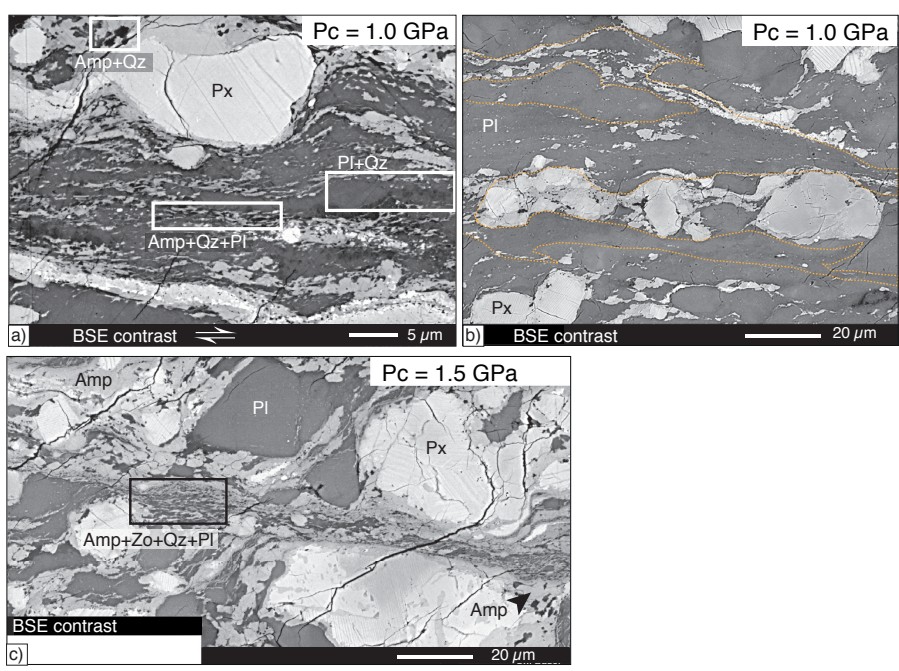

**Figure 6.** Distribution of phases in Maryland Diabase. a) Extensive phase mixing in a shear band: Mixing of Pl+Qz, Pl+Zo, and Amp+Qz. Mixing between Amp and Pl is less frequent. Px clasts show Amp coronas and asymmetric Amp tails. b) Shear bands are predominantly composed of polycrystalline Pl. c) Extensive phase mixing between Amp+Zo+Qz(+Pl) within shear bands. Px clasts show Amp coronas and asymmetric Amp tails.

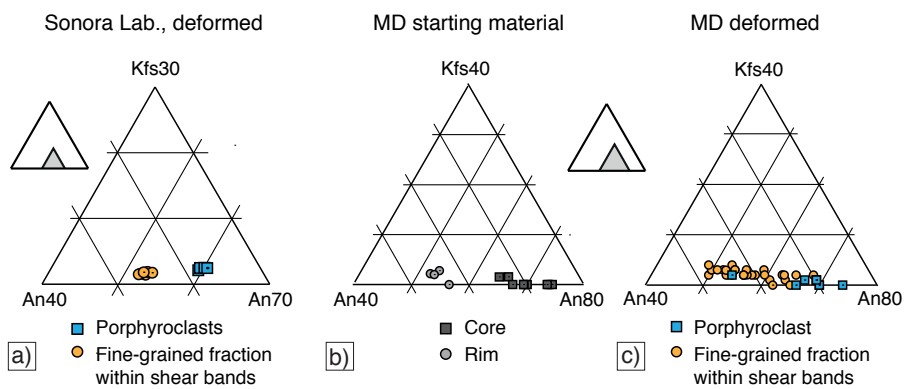

**Figure 7.** Plagioclase chemical compositions. a) Sonora Labradorite in An60+Di experiment runs. b) Maryland Diabase starting material. c) Maryland Diabase after the experiment. Porphyroclasts vs. small new grain fraction found in shear bands.





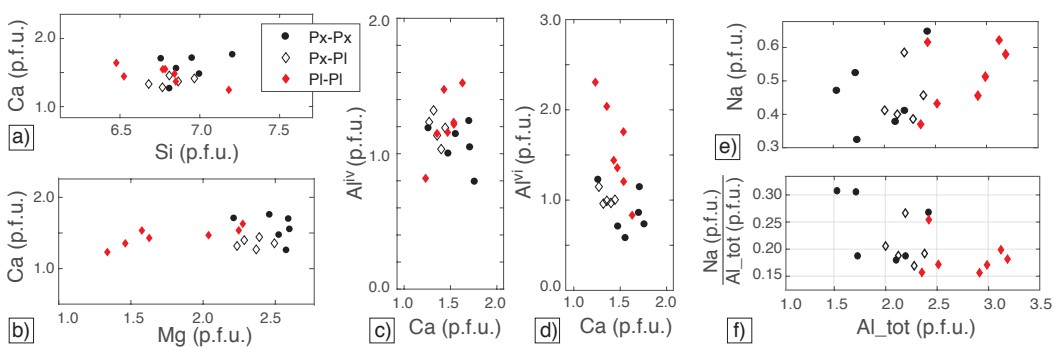

**Figure 8.** Amphibole chemistry. Amphibole grains of Maryland Diabase experiments performed at Pc ≈ 1.0 GPa. Measurements are grouped accoding to their neighbourhood: Px-Px = pyroxene dominated neighbourhood; Px-Pl = Amphibole grown between pyroxene and plagioclase grains; Pl-Pl = plagioclase dominated neighbourhood. a) Ca vs Si per formula unit (p.f.u.). b) Ca vs. Mg. c) estimated $Al^{iv}$ vs. Ca. d) estimated $Al^{vi}$ vs. Ca. e) Na vs. Al (total). f) Na per Al (total) ratio vs. Al (total).





**Table 3.** EDS measurements of Amp chemical compositions from samples deformed at Pc $\approx$ 1.0GPa. Amphibole classification after Hawthorne et al. (2012). Tschermak = Tschermakite, Mg Hornbl = Magnesium Hornblende. All Fe is taken as $Fe^{2+}$ due to the reducing conditions in the sample assembly.

| normalized to 98 % wt.-% | **Amphibole** | | | |
|---|---|---|---|---|
| | 414sm Tschermak. | 414sm Mg Hornbl | 490sm Mg Hornbl | 490sm Tschermak. |
| $SiO_2$ | 45.18 | 45.72 | 47.76 | 47.31 |
| $Al_2O_3$ | 17.13 | 14.13 | 13.06 | 17.73 |
| CaO | 9.24 | 8.92 | 9.45 | 10.02 |
| $Na_2O$ | 1.63 | 1.74 | 2.12 | 1.84 |
| $K_2O$ | 1.26 | 0.86 | 0.89 | 0.90 |
| MgO | 7.56 | 9.95 | 11.24 | 7.39 |
| $TiO_2$ | 0.00 | 1.78 | 0.00 | 0.00 |
| FeO | 15.99 | 14.90 | 13.48 | 12.81 |
| MnO | 0.00 | 0.00 | 0.00 | 0.00 |
| $Cr_2O_3$ | 0.00 | 0.00 | 0.00 | 0.00 |
| **Total:** | **97.99** | **98.00** | **97.99** | **98.00** |
| Formula per 23 oxygen | | | | |
| Si | 6.59 | 6.76 | 6.89 | 6.77 |
| Ti | 0.00 | 0.00 | 0.00 | 0.00 |
| Al | 2.95 | 2.46 | 2.22 | 2.99 |
| $Fe^{3+}$ | 0.00 | 0.00 | 0.00 | 0.00 |
| Cr | 0.00 | 0.00 | 0.00 | 0.00 |
| Mg | 1.65 | 2.19 | 2.42 | 1.58 |
| Ca | 1.45 | 1.41 | 1.46 | 1.54 |
| Mn | 0.00 | 0.00 | 0.00 | 0.00 |
| $Fe^{2+}$ | 1.95 | 1.84 | 1.63 | 1.53 |
| Na | 0.46 | 0.50 | 0.59 | 0.51 |
| K | 0.23 | 0.16 | 0.16 | 0.16 |
| **Total** | **15.28** | **15.34** | **15.38** | **15.08** |



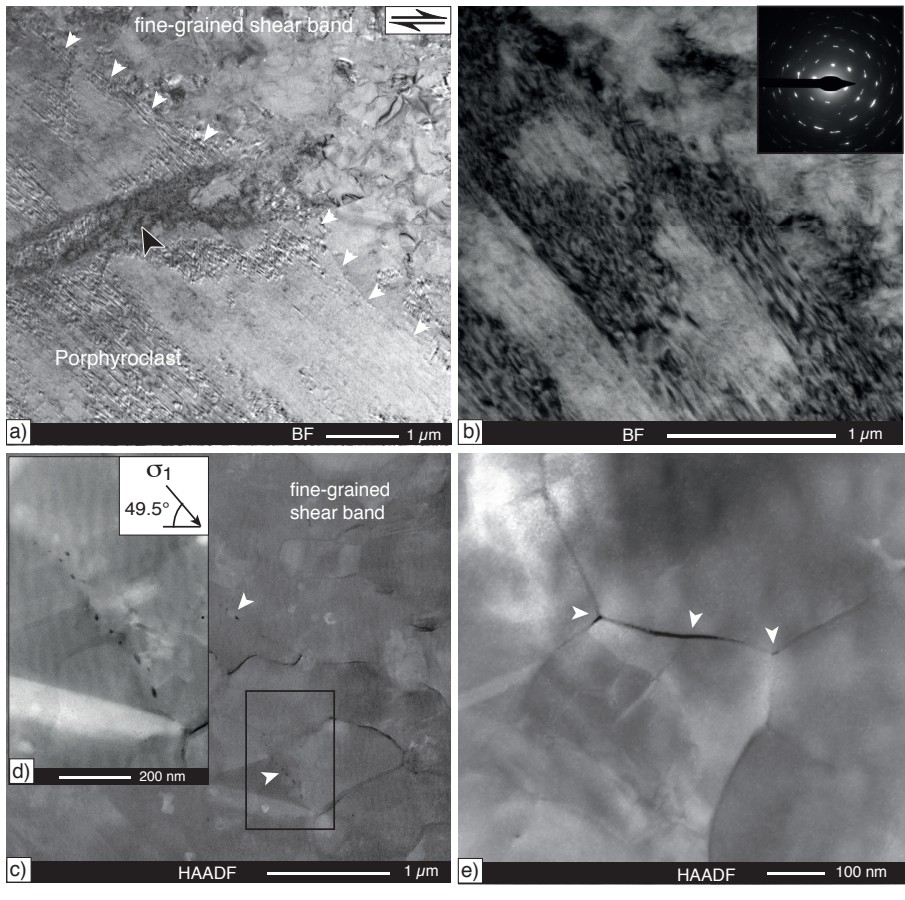

**Figure 9.** Nano-structures of shear bands in Ab+En sample. Sample (518), shear zone boundaries are horizontal, shear sense is dextral. a) Bright-field (BF) image of a plagioclase porphyroclasts adjacent to a fine grained shear band. White arrows mark the porphyroclast-shear band interface. Black arrow points to a high defect density band within the clast. b) BF image of the internal structure of porphyroclast showing high defect density. Twin lamellae run from upper left to lower right. c) HAADF image of a fine-grained plagioclase in a shear band. Black = porosity. White arrows point to open grain boundaries, black arrows indicate pore-trails along grain boundaries. Black rectangle marks close-up view in d). d) HAADF image of a pore-trail following several aligned grain boundaries. The local orientation of $\sigma_1$ is derived from the orientation of the ISA (Table 2). e) HAADF image. White arrows point to porosity or opening sites developed along two triple-junctions and a grain boundary.





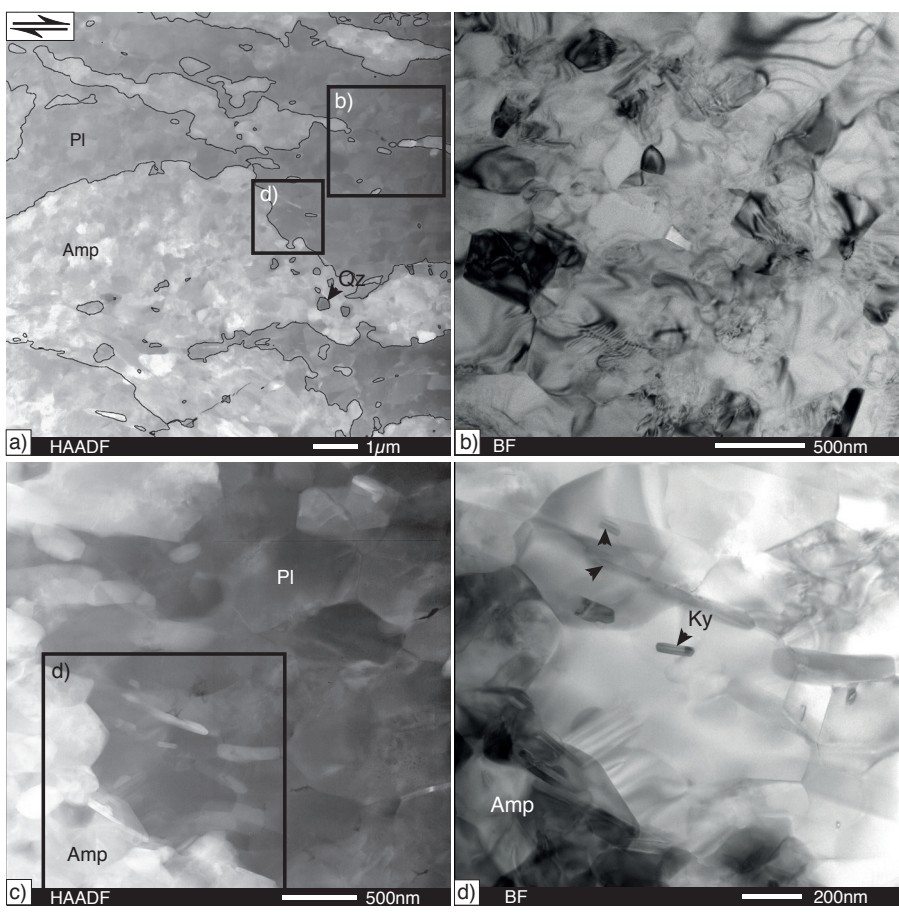

**Figure 10.** Nano structure of shear bands in Maryland Diabase sample. Sample (414), shear zone boundaries are horizontal, shear sense is dextral. a) HAADF image, overview. Amp aggregates are traced with black lines for better visibility. Rectangle indicate areas shown in b) and d). b) BF-TEM image of small (usually ≤ 600 nm) plagioclase grains with low internal defect densities. Grain boundaries are tight and porosity is scarce. c) HAADF image, overview. Black rectangle indicates area shown in d). d) BF-TEM image of a few Ky and Amp grains growing between Pl grains. Size of all phases is a few 100 nm, grains have a low internal defect density.

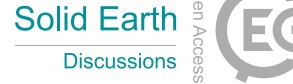

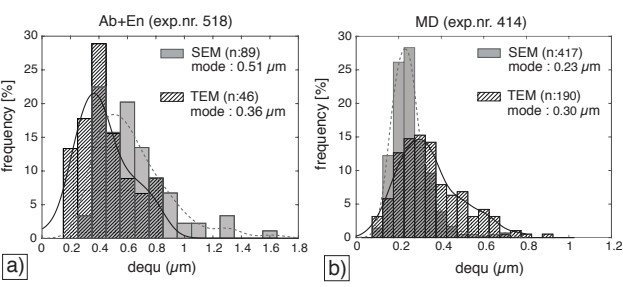

**Figure 11.** Grain size distributions of plagioclase in shear bands. a) Ab+En experiment 518, and b) the Maryland Diabase experiment 414. 2D grain size distributions (GSDs) in both samples are determined on BSE SEM images and TEM images separately. n = number of grains; solid black (TEM) and dashed grey lines (SEM) are kernel density estimate fits to the GSDs and modes determined from the fit are given in the graph.



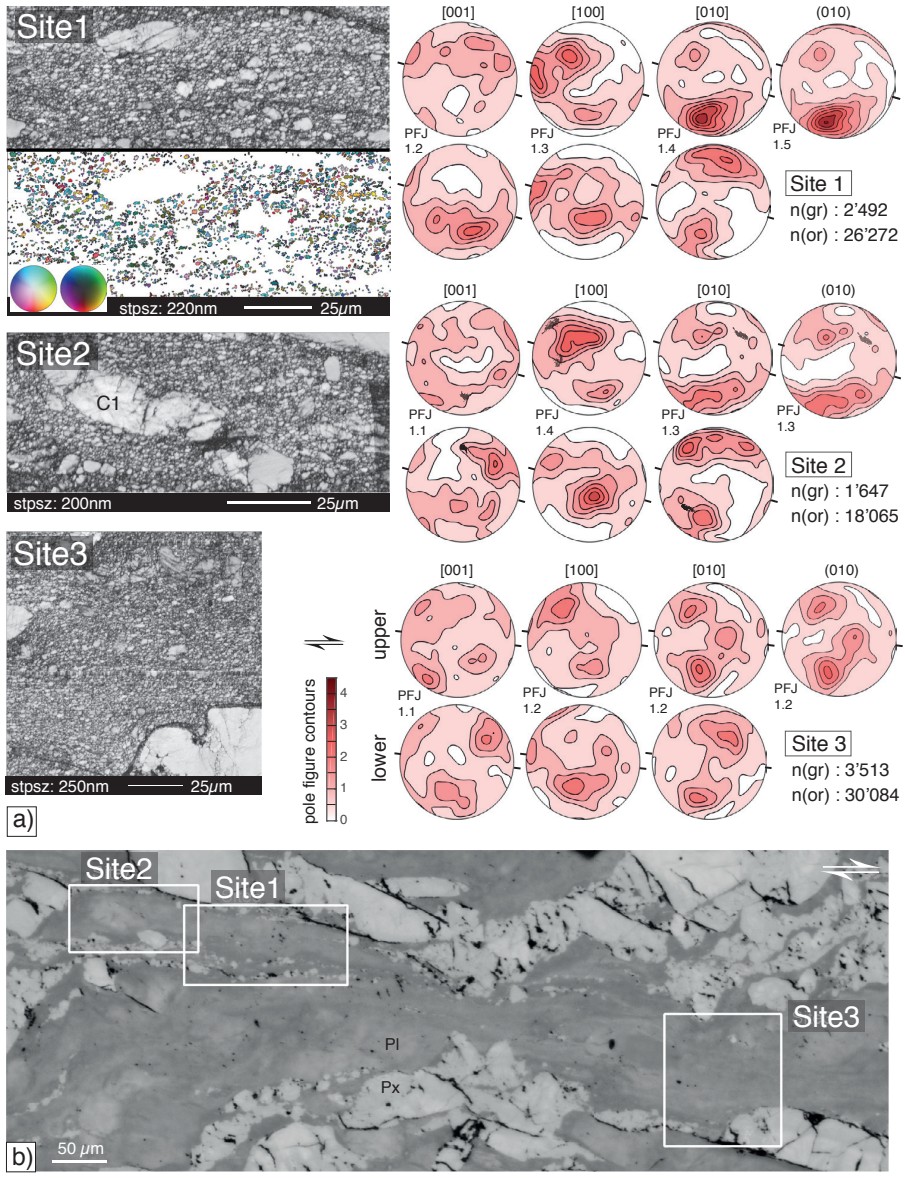

**Figure 12.** a) EBSD analysis of Ab+En sample. a) Band contrast images and contoured pole figures are given for three different sites along one shear band of sample 518 (location of maps are indicated in b). For site 1, the EBSD map (inverse pole figure coloring, x direction) is shown also. Pole figures are upper hemisphere equal angle projections of orientations from grains < 2 $\mu$m in diameter (and > 3 pixels). For site 2, the orientation of the plagioclase porphyroclast (C1) is additionally plotted as poles (black dots) in the pole figures. Contours are at 0.5x multiples of uniform distribution. Black lines at pole figure rims indicate local shear band trend. stpsz = step size of EBSD data acquisition, n(or) = number of EBSD data points used in pole figures, and n(gr) = number of corresponding grains, PFJ = pole figure J-index, indicated at left of pole figures. b) Reflected light image. EBSD map locations are indicated, Pl = plagioclase, Px = pyroxene.



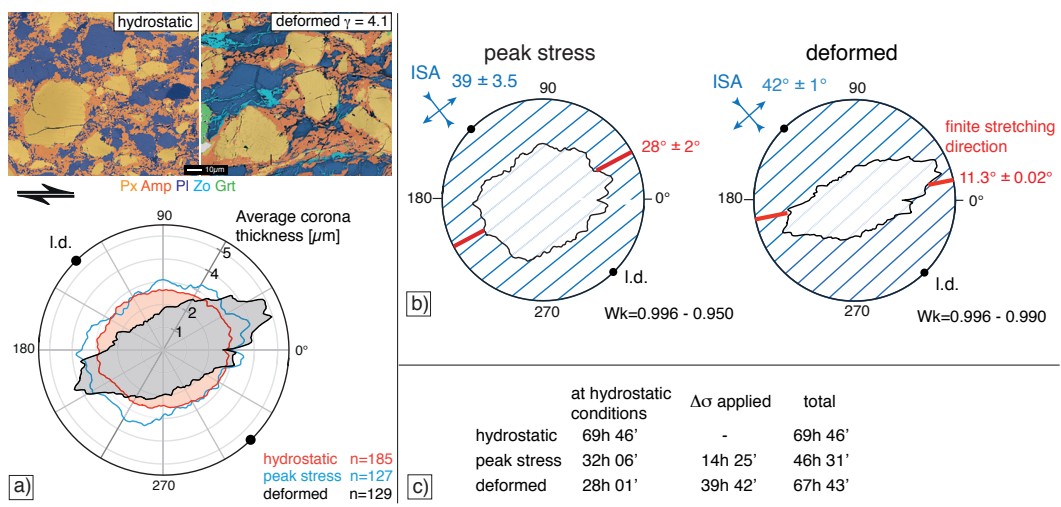

**Figure 13.** Analysis of amphibole coronas. Thickness of amphibole (Amp) corona on pyroxene (Px) porphyroclasts as a function of orientation, from Maryland Diabase experiments deformed at Pc ≈ 1.5 GPa. a) Average Amp corona thickness presented as rose diagram. n = number of analysed coronas. Analysis for three different samples are presented, 'hydrostatic', 'peak stress' ($\gamma_a \sim 0.6$) and 'deformed' ($\gamma_a \sim 4$). b) Duration of experiments in hours (h) and minutes ('). c) Kinematic analysis of a), blue lines indicate calculated instantaneous stretching directions. In red, calculated finite stretching direction. l.d. = loading direction. Wk = kinematic vorticity number. The error range in ISA direction, finite stretching direction and Wk is caused by the uncertainty of the starting thickness of the shear zone.



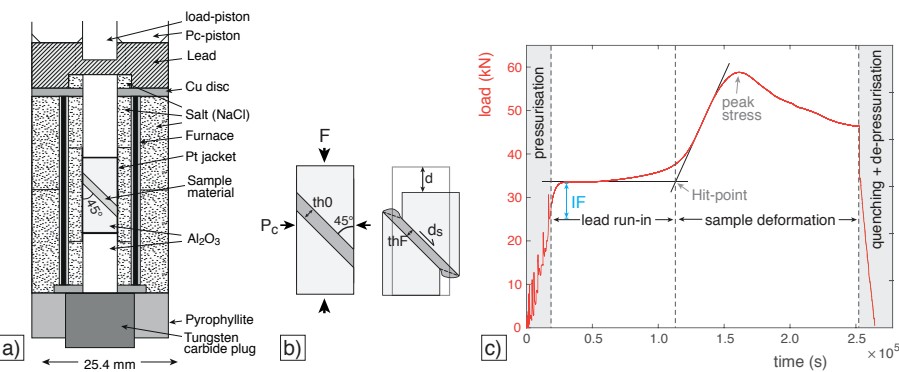

**Figure A.1.** General shear experiments. a) Sample assembly in cross-section. b) Schematic sample cross section at start of experiment (left) and after sample deformation (right). F = load induced by load-piston, Pc = confining pressure, d = axial displacement, ds = shear displacement, th0 = shear zone thickness at start, thF = shear zone thickness at end of experiment. c) Phases of an experiment, red line = load F(t). During 'pressurization', the sample is brought to the desired Pc-T conditions. Black dot denotes start of experiment. IF = initial load increase caused by machine friction. Phase 1: 'lead run-in', the sample not loaded. Phase 2: sample supports load and deforms. 'Hit-point' denotes onset of sample loading. During 'quenching + de-pressurization', the load is released and the temperature brought to ambient conditions.