# Peer review of "Syn-kinematic hydration reactions, dissolution-precipitation creep and grain boundary sliding in experimentally deformed plagioclase - pyroxene mixtures"

_Solid Earth, 2018_

## Referee Comment (RC1) · J. Wheeler (Referee) · 12 Jun 2018

General comments

This paper discusses experiments on deforming pyroxene plagioclase mixtures in the presence of water, a common scenario during metamorphism in the Earth. The work aims to elucidate deformation mechanisms when there are ongoing reactions, by means of mechanical and microstructural observations. The work is of good quality and in general the conclusions are justified but the paper would benefit from some "scene setting". It is not always obvious in advance why various measurements were made, although the interpretations are interesting afterwards. For example coronas

have thickness anisotropy but was that a particular focus of the work, and were there hypotheses to be tested prior to the observations? In terms of results, a bit more discussion would be beneficial. For example two stress exponents, n, are obtained using constant strain rate data and strain rate stepping experiments, but it is not explained why they are different. The observations on porosity are interesting – it is difficult to envisage open pores at 1.5 GPa, but there they are. Is there a chance they formed on sample unloading? The initial powders were highly porous presumably. I can't tell if they were more or less fully compacted prior to deformation. Other points are made below.

Specific comments

1. Abstract first line is vague – what exactly is poorly constrained?

37. diffusion creep always involves GBS; see also 6, 288, and Raj and Ashby (1971) p. 1120. Title: "Syn-kinematic hydration reactions, dissolution-precipitation creep and grain boundary sliding in experimentally deformed plagioclase -pyroxene mixtures" does not really need reference to GBS as it is implicit. 43. "diffusion is expected to be faster along phase boundaries compared to grain boundaries". Wheeler (1992) does not say this, instead he shows that coupled diffusion and reaction may enhance strain along heterophase boundaries even if diffusion coefficients are the same as along single phase boundaries. I think that work has been misquoted previously. I do not know if the other cited works are explicit that diffusion is expected to be faster along phase boundaries compared to grain boundaries. 55, 60. Aims vague 95. overestimates – how? 105. Corona thickness is an interesting property – but, to be clear, why was it measured? What hypothesis was being tested? 107. "separated manually" needs further explanation. 133. "Strain rate stepping tests" – why, what is their role? 141. Reaction R1 is not easily balanced, and will be different for Opx and Cpx. Any comments? 178. "increased amount of reaction products at higher Pc". Any idea why? 287. Why are stress exponents different? 296. Why does cation ordering affect strength (give reference)? 314. "Furthermore, a CPO can form due to interfacial energy, e.g., via

host-controlled nucleation (e.g., ..." (Jiang et al. 2000)). This is a rather confusing phrase. What does "due to" interfacial energy mean? Nucleation rate is certainly influenced by interfacial energy, but surely nucleation is "due to" (in this case) a chemical driving force, namely to change plagioclase to albite. 326. Or could the coronas have been squeezed out somehow? Move line 343 up here to answer. 334. "The geometry of deformation by diffusion creep is irrotational" – please explain. Grains DO rotate during diffusion creep, surely, which is one way it weakens a prior CPO. 399. This seems reasonable but is not entirely logical. Of course finer grain sizes are more prone to DPC but microstructures are always hard to interpret. The best evidence is the low stress exponents. 404. "the chemical driving potential for attaining a lower energy assemblage partially controls the reaction rate". Broadly yes except there is no unique driving force (no Gibbs free energy defined) in a stressed system (Wheeler 2014, 2018) .

Technical corrections

82. Tromso – Tromsø 99. ration – ratio 196. amphibole 260. accompanied by 363. earlier 373. What is t in $\Delta Gt$? Fig 1. What is CCL? Table 1, 3. Poor quality – reformat? Fig. 5k Make colours a bit stronger? Fig 7b) In colour, like others?

Jiang, Z., Prior, D. J. & Wheeler, J. 2000. Albite crystallographic preferred orientation and grain misorientation distribution in a low-grade mylonite: implications for granular flow. Journal of Structural Geology 22, 1663-1674. Raj, R. & Ashby, M. F. 1971. On grain boundary sliding and diffusional creep. Metallurgical Transactions 2, 1113-1127. Wheeler, J. 1992. The importance of pressure solution and Coble creep in the deformation of polymineralic rocks. Journal of Geophysical Research 97, 4579-4586. Wheeler, J. 2014. Dramatic effects of stress on metamorphic reactions. Geology 42(8), 647-650. Wheeler, J. 2018. The effects of stress on reactions in the Earth: sometimes rather mean, usually normal, always important. Journal Of Metamorphic Geology 36, 439-461.

---

## Referee Comment (RC2) · A. Cross (Referee) · 12 Jun 2018

General comments

This paper describes a series of deformation experiments performed on hydrous mixtures of plagioclase and pyroxene, designed to investigate the influence of syn-kinematic reaction on the strength and deformation mechanisms of lower crustal rocks. Through detailed microanalysis, the authors conclude that reaction-driven grain size reduction enhanced dissolution-precipitation creep, leading to strain localization. Overall, this is an important and well-executed piece of work. However, I would like the authors to more thoroughly discuss the evolution of porosity through the experiments. The

starting materials (powders) were hot-pressed in-situ during the PT ramp and run-in. No details are given regarding the porosity of the starting material, and it is possible that significant reaction took place before deformation began, while the samples were not fully densified. Observations of porosity/dilation in the deformed samples imply differential stresses in excess of the confining pressure (1-1.5 GPa), which are not supported by the mechanical data. Nevertheless, with some clarification, this has the potential to be a valuable contribution. The writing and figures are generally of excellent quality, although a few minor clarifications are needed, as detailed below.

Specific comments/corrections

- Line 16 – need to be careful when talking about diffusion creep and grain boundary sliding as separate mechanisms. Grain boundary sliding always occurs during diffusion creep, as an accommodation mechanism for changing grain shapes (see Raj & Ashby, 1971; Gifkins, 1976). - Lines 20-21 – references needed for "It is often suggested that viscous deformation in monomineralic aggregates at mid- to lower crustal conditions is dominated by dislocation creep" - Line 29 – comma needed after "differential stress" - Line 47 – worth pointing out here that, in the absence of fluids/reaction, phase mixing is extremely inefficient (e.g., Linckens et al., 2014; Cross & Skemer, 2017). Thus, strain may preferentially localize into wet/reactive portions of the lithosphere (this is also supported by the experiments performed here, showing extensive phase mixing at low strains). - Line 70 – full-stop/period needs to be removed after "enstatite" - Lines 85-87 – important to mention here that the samples were hot-pressed in-situ at experiment conditions during the run-in to the hit-point. What was the porosity of the starting material at the hit-point in the 1.0 GPa and 1.5 GPa experiments? How much reaction took place during the ramp to PT conditions, and during the run-in to the sample hit-point? - Lines 70-72 – best to add all the abbreviations used here, to match with those given in Table 2 - Line 89 – need to mention that thickness is measured parallel to the shear plane normal - Lines 95-96 – this needs a bit of re-wording. $\gamma$a will underestimate shear strain in localized zones, and overestimate shear strain in undeformed/low-strain

zones. - Line 99 – "strain ratio" instead of "strain ration" - Line 129 – use of the word "near" here is a bit subjective. None of the samples exceed 50-60% of the Goetze criterion. It's probably sufficient to say that none of the samples exceeded the Goetze criterion, so brittle/dilational behavior is not anticipated (presence of open pores contradicts this, however – see below). - Section 3.2 – if phase/element maps are available (like those in Figure 13), these would be preferable for showing the reactions (it's difficult to tell all the phases apart on the BSE images in Figure 5) - Line 204 – I think this should be "intragranular" instead of "intergranular" - Line 207 – the presence of pores contradicts an earlier statement about the Goetze criterion not being exceeded, unless large local stresses along grain boundaries were sustained through the experiments. Alternatively, the opening sites shown in Figure 9 (particularly 9e, for example) could have formed during decompression. - Section 3.4.3 – given the low symmetry of plagioclase (and large number of documented slip planes/directions), it may be more informative to determine slip systems using inverse pole figures – e.g., parallel to the shear direction, perpendicular to shear plane. See Fig. 11 in Miranda et al., 2016, JSG, for example, which shows (011)[-100] as the dominant slip system for intermediate-composition plagioclase (deformed at similar conditions to this study). - Lines 314-315 – I think it's more a case of host-controlled growth. Grains may nucleate in any orientation, but those with low interfacial energies w.r.t. the host will be the ones to grow. - Lines 320-323 – are you able to say anything about the feasibility of the other CPO-forming mechanisms described here? Do grains have a crystallographically-controlled shape; are there systematic interphase misorientation relationships indicative of host-controlled nucleation/ growth? This would be interesting to add, but may be beyond the scope of the paper... - Line 364 – misspelling of "earlier" - Line 373 – G and t need to be defined - Lines 381-389 – I'm not sure what the significance of this paragraph is. Are the authors suggesting that fractures formed in regions of high dislocation densities, leading to enhanced fluid flow and reaction? Again, given that Pc was 2-3 times higher than differential stress, it is difficult to imagine that dilation occurred. - Line 395 – "DPC" instead of "DCP" - Figure 1 caption – use "counterclockwise" instead of CCL

- Figure 5 – it would be useful to point out where the "shear band close-up" images come from in the "overview" images - Figure 5k – the phase map colours are very faint, and are difficult to tell apart

References

Raj, R., & Ashby, M. F. (1971). On grain boundary sliding and diffusional creep. Metallurgical transactions, 2(4), 1113-1127.

Gifkins, R. (1976). Grain-boundary sliding and its accommodation during creep and superplasticity. Metallurgical transactions a, 7(8), 1225-1232.

Linckens, J., Bruijn, R. H., & Skemer, P. (2014). Dynamic recrystallization and phase mixing in experimentally deformed peridotite. Earth and Planetary Science Letters, 388, 134-142.

Cross, A. J., & Skemer, P. (2017). Ultramylonite generation via phase mixing in high-‐strain experiments. Journal of Geophysical Research: Solid Earth, 122(3), 1744-1759.

Miranda, E. A., Hirth, G., & John, B. E. (2016). Microstructural evidence for the transition from dislocation creep to dislocation-accommodated grain boundary sliding in naturally deformed plagioclase. Journal of Structural Geology, 92, 30-45.

---

## Editor Comment (EC1) · M. J. Heap (Editor) · 15 Jun 2018

Dear authors,

As you can see, I've now received two reviews of your manuscript "Syn-kinematic hydration reactions, dissolution-precipitation creep and grain boundary sliding in experimentally deformed plagioclase - pyroxene mixtures". The reviews are generally positive, but both highlight areas in which in the manuscript could be improved. Reviewer #1 asks for some "scene setting" and a deeper discussion of the data, and Reviewer #2 thinks the manuscript would benefit from a more detailed discussion of the porosity evolution. I agree with their assessments. If you are willing, please now prepare a

revised manuscript (with all the changes to the manuscript clearly highlighted) and a point-by-point rebuttal letter.

Thanks for submitting your work to Solid Earth.

Mike Heap (Topical Editor of Solid Earth)

---

## Author Comment (AC1) · 11 Jul 2018

Dear Editor,
We've modified the manuscript in response to the reviewers comments (text changes in the revised manuscript are marked in blue) and hope that it is now in an acceptable form for the publication in Solid Earth. Detailed responses to each reviewer will be submitted but find below a short summary of the more important changes.
With best regards,
Sina Marti (on behalf of the authors)

[Figure]

**Changes to the manuscript**
Changed title to:
*"Syn-kinematic hydration reactions, grain size reduction and dissolution-precipitation creep in experimentally deformed plagioclase - pyroxene mixtures"*
(originally: *"Syn-kinematic hydration reactions, dissolution-precipitation creep and grain boundary sliding in experimentally deformed plagioclase - pyroxene mixtures"*)

Major changes have been made to subsection *"3.4.3 Albite crystallographic preferred orientation"* and *"4.5 Albite crystallographic preferred orientation"* as Figure 12 was changed to include inverse pole figures (upon request from Reviewer 2, Andrew Cross).
Figure 12 has been changed to include inverse pole figures and the previously presented pole figures have been replaced.

Changed order of subsections in the discussion:
Subsection *"Dissolution-precipitation creep and grain boundary sliding"* was moved up to subsection 4.3 (was originally subsection 4.5)
Subsection *"Albite crystallographic preferred orientation"* was moved down to subsection 4.5 (was originally subsection 4.3)
* * *

---

## Author Comment (AC2) · 11 Jul 2018

**Author's response to comments from Reviewer 1 (J.W.)**

*General comments*

*Rev.1: This paper discusses experiments on deforming pyroxene plagioclase mixtures in the presence of water, a common scenario during metamorphism in the Earth. The work aims to elucidate deformation mechanisms when there are ongoing reactions, by*

[Figure]

*means of mechanical and microstructural observations. The work is of good quality and in general the conclusions are justified but the paper would benefit from some "scene setting". It is not always obvious in advance why various measurements were made, although the interpretations are interesting afterwards. For example coronas have thickness anisotropy but was that a particular focus of the work, and were there hypotheses to be tested prior to the observations? In terms of results, a bit more discussion would be beneficial. For example two stress exponents, n, are obtained using constant strain rate data and strain rate stepping experiments, but it is not explained why they are different. The observations on porosity are interesting – it is difficult to envisage open pores at 1.5 GPa, but there they are. Is there a chance they formed on sample unloading? The initial powders were highly porous presumably. I can't tell if they were more or less fully compacted prior to deformation. Other points are made below.*

**Authors:** We would like to thank you for your revisions and greatly appreciate your comments and suggestions. We agree that the manuscript would benefit from some more "scene setting" and tried to incorporate such especially in the sections on EBSD and amphibole corona analyses.

Concerning the pores discussed in our manuscript – they were also commented by Reviewer 2 – this discussion point was added to the specific comments, see below.

Samples were almost fully compacted prior to sample deformation (revised manuscript lines 169-170)

———————————————————

***Specific comments***

*Rev.1: 1. Abstract first line is vague – what exactly is poorly constrained?*

**Authors:** Text altered to *"It is widely observed that mafic rocks are able to accommodate high strains by viscous flow. Yet, a number of questions concerning the exact nature of the involved deformation mechanisms continue to be debated."* (Revised manuscript lines 1 - 2).

————————————————————

*Rev.1: 37. diffusion creep always involves GBS; see also 6, 288, and Raj and Ashby (1971) p. 1120. Title: "Syn-kinematic hydration reactions, dissolution- precipitation creep and grain boundary sliding in experimentally deformed plagioclase -pyroxene mixtures" does not really need reference to GBS as it is implicit.*

**Authors:** Thank you for pointing this out. We are not unaware of the relationship between diffusion creep and GBS. In the original manuscript, we listed both of them separately, as either diffusion creep or GBS could be the dominant strain accommodating mechanism, while the other is merely accommodating (diffusion creep accommodated by GBS vs. GBS accommodated by diffusion creep, e.g. correlating to Lifshitz and Rachinger sliding after Langdon, 2006, respectively).
The manuscript text lines have been modified, see revised manuscript lines 16- 22.
The title has been altered in response to this comment as well.

————————————————————

*Rev.1: 43. "diffusion is expected to be faster along phase boundaries compared to grain boundaries". Wheeler (1992) does not say this, instead he shows that coupled diffusion and reaction may enhance strain along heterophase boundaries even if diffusion coefficients are the same as along single phase boundaries. I think that work*

*has been misquoted previously. I do not know if the other cited works are explicit that diffusion is expected to be faster along phase boundaries compared to grain boundaries.*

**Authors:** True, in the cited literature (Hickman and Evans, 1991; Wheeler, 1992; Sundberg and Cooper, 2008) it is not referred to the diffusion coefficient (i.e. no statement about higher diffusion coefficient along phase- compared to grain- boundaries), but it is observed or suggested that convergence/divergence between two different mineral phases appears faster than between two grains of the same mineral phase. The respective manuscript text passage has been modified, see revised manuscript lines 48 – 51. Do you agree with this reformulation?

———————————————————————

*Rev.1: 95. overestimates – how?*

**Authors:** The way in which $\gamma_a$ is calculated is an incremental approach where the increments are determined by the sampling frequency of the displacement transducer. The end-value of $\gamma_a$ then is a function of the increment size. Comparison of $\gamma_a$ with the simple shear component ($\gamma$) calculated after Fossen & Tikoff, (1993) (Figure attached, from Marti et al., 2017) show that $\gamma_a$ is always a bit higher and is considered to overestimate the simple shear component in the general shear progressive deformation. The text passages have also been criticized by reviewer 2 and were modified, see revised manuscript lines 108 - 114.

———————————————————————

*Rev.1: 105. Corona thickness is an interesting property – but, to be clear, why was it measured? What hypothesis was being tested?*

**Authors:** Some introduction to why the amphibole coronas were measured has been added in the revised manuscript lines 267 – 270, and in lines 349 – 352.

———————————————————

*Rev.1: 107. "separated manually" needs further explanation.*

**Authors:** Some more information/explanation to this has been added in lines 122 – 124.

———————————————————

*Rev.1: 133. "Strain rate stepping tests" – why, what is their role?*

**Authors:** to test the sensitivity of shear stress on strain rate." Revised manuscript line 155.

———————————————————

*Rev.1: 141. Reaction R1 is not easily balanced, and will be different for Opx and Cpx. Any comments?*

**Authors:** Reaction balancing has been attempted and yes, for hornblende as a reaction product needs both Opx and Cpx. These phases are present in the Maryland diabase starting material. As both the measured pyroxene and the resulting amphibole chemistry show some variations, reaction balancing did not yield a single solution and only the schematic reaction (without exact balancing) is listed in the manuscript.

———————————————————

*Rev.1:* *178. "increased amount of reaction products at higher Pc". Any idea why?*

**Authors:** Probably due to the fact that zoisite and albite are favored over intermediate Plagioclase at the higher Pc of 1.5 GPa. From thermodynamic modeling, plagioclase in the Maryland Diabase experiments should, for the most part, disappear by reaction. However, the mineral assemblage of the calculated pseudosections shows some deviation from the observed assemblage in the samples. It is therefore difficult to make assumptions based on these pseudosections.
————————————————————

*Rev.1:* *287. Why are stress exponents different?*

**Authors:** With the database we have, this question is difficult to answer. Explanations could be:
(i) The microstructure in the samples is heterogeneous due to the strain localization. The stress exponents are determined from mechanical data that only measures bulk sample response. Small differences in microstructures such as shear band orientation and shear band interconnectivity might lead to different bulk determined stress exponents.
(ii) A potential switch from interface to transport control of the dissolution- precipitation creep (DPC) rate could potentially explain the difference in stress exponent. The intermediate plagioclase in 1.0 GPa Pc experiments is largely stable, whereas it is highly metastable in the 1.5 GPa Pc experiments. The driving force for dissolution of the intermediate plagioclase and growth of a more stable phase in the 1.5 GPa Pc experiment thus could be expected to be much higher compared to the 1.0 GPa Pc experiments. DPC at 1.5 GPa Pc could be transport controlled, whereas DPC at 1.0

GPa Pc could be interface controlled.

(iii) The data set from which the stress exponents are determined is relatively small. It cannot be fully ruled out, that the small difference in stress exponent is an artifact from too small sampling size (not enough data points to cover the variability between individual experiments)

——————————————————————

*Rev.1: 296. Why does cation ordering affect strength (give reference)?*

**Authors:** The part has been deleted from the sentence. See revised manuscript line 377 - 379.

——————————————————————

*Rev.1: 314. "Furthermore, a CPO can form due to interfacial energy, e.g., via host-controlled nucleation (e.g., .." (Jiang et al. 2000)). This is a rather confusing phrase. What does "due to" interfacial energy mean? Nucleation rate is certainly influenced by interfacial energy, but surely nucleation is "due to" (in this case) a chemical driving force, namely to change plagioclase to albite.*

**Authors:** Whereas the dominant driving force for nucleation is likely to be a chemical driving force, the crystal orientation of the growing nucleus could be determined by the neighboring crystals. That is, the grain boundary energy between nuclei and the neighboring grains may favour a specific orientation, if a certain orientation between matrix grain and nucleus has a particularly low energy. If the matrix grains show a CPO, this could then cause a CPO of the newly nucleating grains.

The manuscript text has been slightly altered regarding this, see revised manuscript lines 394 - 396.

———————————————

*Rev.1:* 326. Or could the coronas have been squeezed out somehow? Move line 343
up here to answer.

**Authors:** Lines 343f (original manuscript) were moved up to revised manuscript
lines 354 - 357. It is stated there that based on the microstructure, squeezing out or
shearing off is not seen as a likely cause for the amphibole corona thickness decrease
in high stress sites.

———————————————

*Rev.1:* 334. "The geometry of deformation by diffusion creep is irrotational" – please
explain. Grains DO rotate during diffusion creep, surely, which is one way it weakens
a prior CPO.

**Authors:** True, the sentence needs rewording (see revised manuscript lines 366 -
369). What we wanted to point out is that the instantaneous grain shape change
expected to result from DPC would be orthorhombic, as a grain gets dissolved and
grows with respect to the orthogonal stress field (i.e. dissolving towards sigma1 and
growing towards sigma3). As soon as any grain boundary sliding is involved (i.e.
during progressive deformation), grains are very likely to rotate, which will likely also
cause grain shape to deviate from an orthogonal shape. We think that this is nicely
shown in the corona evolution, where at very low strains (at peak stress, gamma
smaller equal 1) the average corona shape shows this orthogonal shape, where at
high strains, this shape fabric changed to a monoclinic/sigmoidal shape.

———————————————

*Rev.1: 399. This seems reasonable but is not entirely logical. Of course finer grain sizes are more prone to DPC but microstructures are always hard to interpret. The best evidence is the low stress exponents.*

**Authors:** The sentence has been deleted from the text.
————————————————————————

*Rev.1: 404. "the chemical driving potential for attaining a lower energy assemblage partially controls the reaction rate". Broadly yes except there is no unique driving force (no Gibbs free energy defined) in a stressed system (Wheeler 2014, 2018).*

**Authors:** We agree, thank you for pointing this out, the text has been modified and citations added, see revised manuscript lines 457-458.
————————————————————————

*Rev.1: The observations on porosity are interesting – it is difficult to envisage open pores at 1.5 GPa, but there they are. Is there a chance they formed on sample unloading? The initial powders were highly porous presumably. I can't tell if they were more or less fully compacted prior to deformation.*

**Authors:** We disagree that the presence of pores is contradicting with the experimental conditions. Surely the high confining pressures and the activation of viscous deformation in the material will suppress large amounts of pore space opening, however this should not mean that no porosity at all is able to exist.
In fact, the observation of (small amounts of) porosity in experiments performed at high Pc & T conditions (with flow stresses below the Goetze criterion) has previously been observed in a number of studies, e.g. Tullis & Yund (1991); Dimanov et al.

(2007); Rybacki & Dresen (2010); Precigout & Stünitz (2016). And is as well proposed for natural shear zones, e.g. Fusseis et al. (2009); Menegon et al. (2015).

A reason why porosity in experiments is not that uncommon to observe might be the high strain rates – in natural rocks, dilatant sites during grain boundary sliding can be filled by precipitating phases (e.g. Kruse & Stünitz, 1999; Kilian et al., 2011) or closed by plastic deformation of adjacent grains. However, as the displacement rate in experiments is high, pores might be more frequent to form as reaction rate and plastic deformation are not able to keep up with pore space formation.

The pores in our experiments are not very frequent and are also small, with sizes on the 10x nm scale. We don't know how long they are open but it is likely that their occurrence time is short.

Decompression porosity commonly is easily recognized by its location and orientation in cracks normal to the shortening direction – such features are different from what is described here.

The samples are almost fully compacted after the lead run-in (hydrostatic part of the experiment, see Appendix Figure 1c, prior to sample deformation)
* * *
**Technical corrections**

*Rev.1: 82. Tromso – Tromsø 99. ration – ratio 196. amphibole 260. accompanied by 363. earlier*
**Authors:** Thank you for pointing this out, the text has been corrected accordingly

*Rev.1: 373. What is t in $\triangle Gt$?*
**Authors:** $\triangle$Gt has been removed as it is not further used in the text.

**Rev.1:** *Fig 1. What is CCL?*
**Authors:** CCL is a typo, is now corrected and replaced with "counter clockwise"

**Rev.1:** *Table 1, 3. Poor quality – reformat?*
**Authors:** Table 1 and 3 are reformatted

**Rev.1:** *Fig. 5k Make colours a bit stronger?*
**Authors:** Colours in Figure 5k have been enhanced.

**Rev.1:** *Fig 7b) In colour, like others?*
**Authors:** Colours have been modified

Rybacki, E., Wirth, R. and Dresen, G., 2010. Superplasticity and ductile fracture of synthetic feldspar deformed to large strain. Journal of Geophysical Research, 115. B08209. doi:10.1029/2009JB007203

Dimanov, A., Rybacki, E., Wirth, R. and Dresen, G., 2007. Creep and strain-dependent microstructures of synthetic anorthite-diopside aggregates. Journal of Structural Geology, 29. 1049-1069.

Tullis, J. and Yund, R. A., 1991. Diffusion creep in feldspar aggregates: experimental evidence. Journal of Structural Geology, 13(9). 987-1000.

Precigout, J. and Stünitz, H., 2016. Evidence of phase nucleation during olivine diffusion creep: A new perspective for mantle strain localization. EPSL, 455. 94-105.

Fusseis, F., Regenauer-Lieb, K., Liu, J., Hough, R.M., and De Carlo, F., 2009, Creep cavitation can establish a dynamic granular fluid pump in ductile shear zones: Nature, v. 459, p. 974–977, doi:10.1038/nature08051.

Menegon, L, Fusseis, F., Stünitz, H., and Xiao X., 2015. Creep cavitation bands control porosity and fluid flow in lower crustal shear zones. Geology. doi: 10.1130/G36307.1

Kruse, R. and Stünitz, H.: Deformation mechanisms and phase distribution in mafic high-temperature mylonites from the Jotun Nappe, southern Norway, Tectonophysics, 303, 223–249, 1999.

Kilian, R., Heilbronner, R., and Stünitz, H.: Quartz grain size reduction in a granitoid rock and the transition from dislocation to diffusion creep, Journal of Structural Geology, 33, 1265 – 1284, 2011.

[Figure]

**Fig. 1.** Appendix Fig. 3 (Marti et al., 2017)

---

## Author Comment (AC3) · 11 Jul 2018

**Author's response to comments from Reviewer 2 (A.C.)**

*General comments*

*Rev.2: This paper describes a series of deformation experiments performed on hydrous mixtures of plagioclase and pyroxene, designed to investigate the influence of syn- kinematic reaction on the strength and deformation mechanisms of lower crustal*

[Figure]

*rocks. Through detailed microanalysis, the authors conclude that reaction-driven grain size re- duction enhanced dissolution-precipitation creep, leading to strain localization. Overall, this is an important and well-executed piece of work. However, I would like the authors to more thoroughly discuss the evolution of porosity through the experiments. The starting materials (powders) were hot-pressed in-situ during the PT ramp and run-in. No details are given regarding the porosity of the starting material, and it is possible that significant reaction took place before deformation began, while the samples were not fully densified. Observations of porosity/dilation in the deformed samples imply differential stresses in excess of the confining pressure (1-1.5 GPa), which are not supported by the mechanical data. Nevertheless, with some clarification, this has the potential to be a valuable contribution. The writing and figures are generally of excellent quality, although a few minor clarifications are needed, as detailed below.*

**Authors:** We would like to thank you for your thorough revisions and greatly appreciate your comments and suggestions. Please find our replies below:

———————————————————

***Specific comments / corrections***

***Rev.2:*** *Line 16 – need to be careful when talking about diffusion creep and grain boundary sliding as separate mechanisms. Grain boundary sliding always occurs during diffusion creep, as an accommodation mechanism for changing grain shapes (see Raj & Ashby, 1971; Gifkins, 1976).*

**Authors:** That is correct, we agree with this statement. The same has been noted

by Reviewer 1 to whom we answered: Thank you for pointing this out. We are not unaware of the relationship between diffusion creep and GBS. In the original manuscript, we listed both of them separately, as either diffusion creep or GBS could be the dominant strain accommodating mechanism, while the other is merely accommodating (diffusion creep accommodated by GBS vs. GBS accommodated by diffusion creep, e.g. correlating to Lifshitz and Rachinger sliding after Langdon, 2006, respectively).

The manuscript text lines have been modified, see revised manuscript lines 16-22. The title has been altered in response to this comment as well.

——————————————————

*Rev.2: Lines 20-21 – references needed for "It is often suggested that viscous defor-mation in monomineralic aggregates at mid- to lower crustal conditions is dominated by dislocation creep"*

**Authors:** The text lines have been modified, see revised manuscript lines 26 – 29.

——————————————————

*Rev.2: Line 47 – worth pointing out here that, in the absence of fluids/reaction, phase mixing is extremely inefficient (e.g., Linckens et al., 2014; Cross & Skemer, 2017). Thus, strain may preferentially localize into wet/reactive portions of the lithosphere (this is also supported by the experiments performed here, showing extensive phase mixing at low strains).*

**Authors:** This is a very important point, thank you for pointing us towards this. It has been incorporated in the revised manuscript lines 61-66.

——————————————————

*Rev.2:* *Line 70 – full-stop/period needs to be removed after "enstatite"*
**Authors:**Redundant period has been removed.
————————————————————

*Rev.2:* *Lines 85-87 – important to mention here that the samples were hot-pressed in-situ at experiment conditions during the run-in to the hit-point. What was the porosity of the starting material at the hit-point in the 1.0 GPa and 1.5 GPa experiments? How much reaction took place during the ramp to PT conditions, and during the run-in to the sample hit- point?*

**Authors:** Notion of the initial hydrostatic stage (lead run-in) has been added in revised manuscript lines 134–137. Amounts of reaction product present in different experiments at different stages can be found in lines 170-175, and notion of initial sample porosity prior to deformation was made in lines 169-170.
————————————————————

*Rev.2:* *Lines 70-72 – best to add all the abbreviations used here, to match with those given in Table 2*

**Authors:**The abbreviations are not included in these lines as they are not further being used in the text and only appear in Table 2 where they are declared in the caption.
————————————————————

*Rev.2:* *Line 89 – need to mention that thickness is measured parallel to the shear plane normal*

**Authors:** Thank you for pointing this out, the text has been modified accordingly. (see revised manuscript line 102)

———————————————————

*Rev.2:* Lines 95-96 – this needs a bit of re-wording. $\gamma_a$ will underestimate shear strain in localized zones, and overestimate shear strain in undeformed/low-strain zones.

**Authors:** The text has been modified accordingly, see revised manuscript lines 108-114.

———————————————————

*Rev.2:* Line 99 – "strain ratio" instead of "strain ration"

**Authors:** Thank you this has been corrected.

———————————————————

*Rev.2:* Line 129 – use of the word "near" here is a bit subjective. None of the samples exceed 50-60% of the Goetze criterion. It's probably sufficient to say that none of the samples exceeded the Goetze criterion, so brittle/dilational behavior is not anticipated (presence of open pores contradicts this, however – see below).

**Authors:** We use the differential stress ($\Delta\sigma$) between the Pc and the load piston to compare with the Goetze criterion not the shear stress $\tau$. $\tau$ supported by the sample inclined at 45° are obtained by Mohr circle construction from $\Delta\sigma$ and are half as much as the $\Delta\sigma$ (see also Appendix A3, lines 531f).

———————————————————

*Rev.2:* Line 204 – I think this should be "intragranular" instead of "intergranular"

**Authors:** We agree, thank you for pointing this out. The text has been modified accordingly.
—————————————————————

*Rev.2:* Line 207 – the presence of pores contradicts an earlier statement about the Goetze criterion not being exceeded, unless large local stresses along grain boundaries were sustained through the experiments. Alternatively, the opening sites shown in Figure 9 (particularly 9e, for example) could have formed during decompression.

**Authors:** We disagree that the presence of pores is contradicting with the experimental conditions. Surely the high confining pressures and the activation of viscous deformation in the material will suppress large amounts of pore space opening, however this should not mean that no porosity at all is able to exist.
In fact, the observation of (small amounts of) porosity in experiments performed at high Pc & T conditions (with flow stresses below the Goetze criterion) has previously been observed in a number of studies, e.g. Tullis & Yund (1991); Dimanov et al. (2007); Rybacki & Dresen (2010); Precigout & Stünitz (2016). And is as well proposed for natural shear zones, e.g. Fusseis et al. (2009); Menegon et al. (2015).
A reason why porosity in experiments is not that uncommon to observe might be the high strain rates – in natural rocks, dilatant sites during grain boundary sliding can be filled by precipitating phases (e.g. Kruse & Stünitz, 1999; Kilian et al., 2011) or closed by plastic deformation of adjacent grains. However, as the displacement rate in experiments is high, pores might be more frequent to form as reaction rate and plastic deformation are not able to keep up with pore space formation.

The pores in our experiments are not very frequent and are also small, with sizes on the 10x nm scale. We don't know how long they are open but it is likely that their occurrence time is short.

Decompression porosity commonly is easily recognized by its location and orientation in cracks normal to the shortening direction – such features are different from what is described here.

The samples are almost fully compacted after the lead run-in (hydrostatic part of the experiment, see Appendix Figure 1c, prior to sample deformation)

————————————————————————

*Rev.2: Section 3.4.3 – given the low symmetry of plagioclase (and large number of documented slip planes/directions), it may be more informative to determine slip systems using inverse pole figures – e.g., parallel to the shear direction, perpendicular to shear plane. See Fig. 11 in Miranda et al., 2016, JSG, for example, which shows (011)[-100] as the dominant slip system for intermediate- composition plagioclase (deformed at similar conditions to this study).*

**Authors:** We have modified Figure 12 to incorporate inverse pole figures for the three different sites. The normal pole figures are reduced to one set of pole figures that combine the data of all three sites together to show the texture pattern. See revised manuscript new Figure 12 and new text passages in sections 3.4.3 (lines 250f) and 4.5 (lines 402f)

————————————————————————

*Rev.2: Lines 314-315 – I think it's more a case of host-controlled growth. Grains may nucleate in any orientation, but those with low interfacial energies w.r.t. the host will be the ones to grow.*

**Authors:** The area covered by the three EBSD maps would include a number of host grains (i.e. now replaced by fine grained albite). If the measured weak CPO is a result of host controlled growth then this would imply a CPO of the initial host grains. We consider this unlikely as e.g. we do not observe any significant amount of dislocation climb or creep in the remaining porphyroclasts.

——————————————————————

*Rev.2:* Lines 320-323 – are you able to say anything about the feasibility of the other CPO- forming mechanisms described here? Do grains have a crystallographically-controlled shape; are there systematic interphase misorientation relationships indicative of host- controlled nucleation/ growth? This would be interesting to add, but may be beyond the scope of the paper. . .

**Authors:** Sadly we do not. The acquired EBSD data does not allow for these analyses, as e.g. (i) the albite grain size is so small that the EBSD points within a single grain are not enough to perform proper shape analyses and (ii) indexing was very low towards grain boundaries or in areas of very fine grains.

——————————————————————

*Rev.2:* Line 373 – G and t need to be defined
**Authors:** $\Delta Gt$ has been removed as it is not further used in the text.

*Rev.2:* Line 364 – misspelling of "earlier"
*Rev.2:* Line 395 – "DPC" instead of "DCP"
*Rev.2:* Figure 1 caption – use "counterclockwise" instead of CCL
**Authors:** Thank you for pointing this out, the text has been corrected accordingly

——————————————————————

*Rev.2:* *Figure 5 – it would be useful to point out where the "shear band close-up"
images come from in the "overview" images*

**Authors:** The close-ups are not within the area of the overview images. We agree that
it would be useful to have close-ups within the overview area. However the images
were selected such that they show the representative microstructure with the best
SEM acquisition quality. Unfortunately when selecting the images according to these
criteria, the close-ups and overview images do not overlap.
————————————————————

*Rev.2:* *Figure 5k – the phase map colours are very faint, and are difficult to tell apart
References*

**Authors:** Colours in Figure 5k have been enhanced.
————————————————————

Rybacki, E., Wirth, R. and Dresen, G., 2010. Superplasticity and ductile fracture of
synthetic feldspar deformed to large strain. Journal of Geophysical Research, 115.
B08209. doi:10.1029/2009JB007203

Dimanov, A., Rybacki, E., Wirth, R. and Dresen, G., 2007. Creep and strain-dependent
microstructures of synthetic anorthite-diopside aggregates. Journal of Structural Ge-
ology, 29. 1049-1069.

Tullis, J. and Yund, R. A., 1991. Diffusion creep in feldspar aggregates: experimental

evidence. Journal of Structural Geology, 13(9). 987-1000.

Precigout, J. and Stünitz, H., 2016. Evidence of phase nucleation during olivine diffusion creep: A new perspective for mantle strain localization. EPSL, 455. 94-105.

Fusseis, F., Regenauer-Lieb, K., Liu, J., Hough, R.M., and De Carlo, F., 2009, Creep cavitation can establish a dynamic granular fluid pump in ductile shear zones: Nature, v. 459, p. 974–977, doi:10.1038/nature08051.

Menegon, L, Fusseis, F., Stünitz, H., and Xiao X., 2015. Creep cavitation bands control porosity and fluid flow in lower crustal shear zones. Geology. doi: 10.1130/G36307.1

Kruse, R. and Stünitz, H.: Deformation mechanisms and phase distribution in mafic high-temperature mylonites from the Jotun Nappe, southern Norway, Tectonophysics, 303, 223–249, 1999.

Kilian, R., Heilbronner, R., and Stünitz, H.: Quartz grain size reduction in a granitoid rock and the transition from dislocation to diffusion creep, Journal of Structural Geology, 33, 1265 – 1284, 2011.